# Dual Adaptivity: A Universal Algorithm for Minimizing the Adaptive Regret of Convex Functions

**Lijun Zhang**[*], **Guanghui Wang**[*], **Wei-Wei Tu**[†], **Wei Jiang**[*], **Zhi-Hua Zhou**[*]

[*]National Key Laboratory for Novel Software Technology, Nanjing University, Nanjing, China
[†]4Paradigm Inc., Beijing 100000, China
{zhanglj, wanggh, jiangw, zhouzh}@lamda.nju.edu.cn, tuwwcn@gmail.com

## Abstract

To deal with changing environments, a new performance measure—adaptive regret, defined as the maximum static regret over any interval, was proposed in online learning. Under the setting of online convex optimization, several algorithms have been successfully developed to minimize the adaptive regret. However, existing algorithms lack universality in the sense that they can only handle one type of convex functions and need apriori knowledge of parameters. By contrast, there exist universal algorithms, such as MetaGrad, that attain optimal static regret for multiple types of convex functions simultaneously. Along this line of research, this paper presents the *first* universal algorithm for minimizing the adaptive regret of convex functions. Specifically, we borrow the idea of maintaining multiple learning rates in MetaGrad to handle the uncertainty of functions, and utilize the technique of sleeping experts to capture changing environments. In this way, our algorithm automatically adapts to the property of functions (convex, exponentially concave, or strongly convex), as well as the nature of environments (stationary or changing). As a by product, it also allows the type of functions to switch between rounds.

## 1   Introduction

Online learning aims to make a sequence of accurate decisions given knowledge of answers to previous tasks and possibly additional information [Shalev-Shwartz, 2011]. It is performed in a sequence of consecutive rounds, where at round $t$ the learner is asked to select a decision $\mathbf{w}_t$ from a domain $\Omega$. After submitting the answer, a loss function $f_t : \Omega \mapsto \mathbb{R}$ is revealed and the learner suffers a loss $f_t(\mathbf{w}_t)$. The standard performance measure is the regret [Cesa-Bianchi and Lugosi, 2006]:

$$\text{Regret}(T) = \sum_{t=1}^{T} f_t(\mathbf{w}_t) - \min_{\mathbf{w} \in \Omega} \sum_{t=1}^{T} f_t(\mathbf{w})$$

defined as the difference between the cumulative loss of the online learner and that of the best decision chosen in hindsight. When both the domain $\Omega$ and the loss $f_t(\cdot)$ are convex, it becomes online convex optimization (OCO) [Zinkevich, 2003].

In the literature, there exist plenty of algorithms to minimize the regret under the setting of OCO [Hazan, 2016]. However, when the environment undergoes many changes, regret may not be the best measure of performance. That is because regret chooses a fixed comparator, and for the same reason, it is also referred to as *static* regret. To address this limitation, Hazan and Seshadri [2007] introduce the concept of adaptive regret, which measures the performance with respect to a changing comparator. Following the terminology of Daniely et al. [2015], we define the strongly adaptive

regret as the maximum static regret over intervals of length $\tau$, i.e.,

$$\text{SA-Regret}(T, \tau) = \max_{[p, p+\tau-1] \subseteq [T]} \left( \sum_{t=p}^{p+\tau-1} f_t(\mathbf{w}_t) - \min_{\mathbf{w} \in \Omega} \sum_{t=p}^{p+\tau-1} f_t(\mathbf{w}) \right). \quad (1)$$

Since the seminal work of Hazan and Seshadhri [2007], several algorithms have been successfully developed to minimize the adaptive regret of convex functions, including general convex, exponentially concave (abbr. exp-concave) and strongly convex functions [Hazan and Seshadhri, 2009, Jun et al., 2017a, Zhang et al., 2018b]. However, existing methods can only handle one type of convex functions. Furthermore, when facing exp-concave and strongly convex functions, they need to know the moduli of exp-concavity and strong convexity. The lack of universality hinders their applications to real-world problems.

On the other hand, there do exist universal algorithms, such as MetaGrad [van Erven and Koolen, 2016], that attain optimal static regret for multiple types of convex functions simultaneously. This observation motivates us to ask whether it is possible to design a single algorithm to minimize the adaptive regret of multiple types of functions. This is very challenging because the algorithm needs to enjoy dual adaptivity, adaptive to the function type and adaptive to the environment. In this paper, we provide an affirmative answer by developing a Universal algorithm for Minimizing the Adaptive regret (UMA). First, inspired by MetaGrad, UMA maintains multiple learning rates to handle the uncertainty of functions. In this way, it supports multiple types of functions simultaneously and identifies the best learning rate automatically. Second, following existing studies on adaptive regret, UMA deploys sleeping experts [Freund et al., 1997] to minimize the regret over any interval, and thus achieves a small adaptive regret and captures the changing environment.

The main advantage of UMA is that it attains second-order regret bounds over any interval. As a result, it can minimize the adaptive regret of general convex functions, and automatically take advantage of easier functions whenever possible. Specifically, UMA attains $O(\sqrt{\tau \log T})$, $O(\frac{d}{\alpha} \log \tau \log T)$ and $O(\frac{1}{\lambda} \log \tau \log T)$ strongly adaptive regrets for general convex, $\alpha$-exp-concave and $\lambda$-strongly convex functions respectively, where $d$ is the dimensionality. All of these bounds match the state-of-the-art results on adaptive regret [Jun et al., 2017a, Zhang et al., 2018b] exactly. Furthermore, UMA can also handle the case that the type of functions changes between rounds. For example, suppose the online functions are general convex during interval $I_1$, then become $\alpha$-exp-concave in $I_2$, and finally switch to $\lambda$-strongly convex in $I_3$. When facing this function sequence, UMA achieves $O(\sqrt{|I_1| \log T})$, $O(\frac{d}{\alpha} \log |I_2| \log T)$ and $O(\frac{1}{\lambda} \log |I_3| \log T)$ regrets over intervals $I_1$, $I_2$ and $I_3$, respectively.

## 2 Related work

We briefly review related work on static regret and adaptive regret, under the setting of OCO.

### 2.1 Static regret

To minimize the static regret of general convex functions, online gradient descent (OGD) with step size $\eta_t = O(1/\sqrt{t})$ achieves an $O(\sqrt{T})$ bound [Zinkevich, 2003]. If all the online functions are $\lambda$-strongly convex, OGD with step size $\eta_t = O(1/[\lambda t])$ attains an $O(\frac{1}{\lambda} \log T)$ bound [Shalev-Shwartz et al., 2007]. When the functions are $\alpha$-exp-concave, online Newton step (ONS), with knowledge of $\alpha$, enjoys an $O(\frac{d}{\alpha} \log T)$ bound, where $d$ is the dimensionality [Hazan et al., 2007]. These regret bounds are minimax optimal for the corresponding type of functions [Abernethy et al., 2008], but choosing the optimal algorithm for a specific problem requires domain knowledge.

The study of universal algorithms for OCO stems from the adaptive online gradient descent (AOGD) [Bartlett et al., 2008] and its proximal extension [Do et al., 2009]. The key idea of AOGD is to add a quadratic regularization term to the loss. Bartlett et al. [2008] demonstrate that AOGD is able to interpolate between the $O(\sqrt{T})$ bound of general convex functions and the $O(\log T)$ bound of strongly convex functions. Furthermore, it allows the online function to switch between general convex and strongly convex. However, AOGD has two restrictions:

- It needs to calculate the modulus of strong convexity on the fly, which is a nontrivial task;
- It does not support exp-concave functions explicitly, and thus can only achieve a suboptimal $O(\sqrt{T})$ regret for this type of functions.

Another milestone is the multiple eta gradient algorithm (MetaGrad) [van Erven and Koolen, 2016, Mhammedi et al., 2019, van Erven et al., 2021], which adapts to a much broader class of functions, including convex and exp-concave functions. MetaGrad's main feature is that it simultaneously considers multiple learning rates and does not need to know the modulus of exp-concavity. MetaGrad achieves $O(\sqrt{T \log \log T})$ and $O(\frac{d}{\alpha} \log T)$ regret bounds for general convex and $\alpha$-exp-concave functions, respectively. However, it suffers the following two limitations:

- MetaGrad treats strongly convex functions as exp-concave, and thus only gives a suboptimal $O(\frac{d}{\lambda} \log T)$ regret for $\lambda$-strongly convex functions;
- It assumes the type of online functions, as well as the associated parameter, does not change between rounds.

The first limitation of MetaGrad has been addressed by Wang et al. [2019], who develop a universal algorithm named as multiple sub-algorithms and learning rates (Maler). It attains $O(\sqrt{T})$, $O(\frac{d}{\alpha} \log T)$ and $O(\frac{1}{\lambda} \log T)$ regret bounds for general convex, $\alpha$-exp-concave, and $\lambda$-strongly convex functions, respectively. Furthermore, Wang et al. [2020] extend Maler to make use of smoothness. However, the second limitation remains there.

## 2.2 Adaptive regret

Adaptive regret has been studied in the setting of prediction with expert advice [Littlestone and Warmuth, 1994, Freund et al., 1997, Adamskiy et al., 2012, György et al., 2012, Luo and Schapire, 2015] and OCO [Hazan and Seshadhri, 2007, Daniely et al., 2015, Jun et al., 2017a]. In this section, we focus on the related work in the latter one.

Adaptive regret is introduced by Hazan and Seshadhri [2007], and later refined by Daniely et al. [2015]. To distinguish between them, we refer to the definition of Hazan and Seshadhri as weakly adaptive regret:

$$\text{WA-Regret}(T) = \max_{[p,q] \subseteq [T]} \left( \sum_{t=p}^{q} f_t(\mathbf{w}_t) - \min_{\mathbf{w} \in \Omega} \sum_{t=p}^{q} f_t(\mathbf{w}) \right).$$

For $\alpha$-exp-concave functions, Hazan and Seshadhri [2007] propose an adaptive algorithm named as Follow-the-Leading-History (FLH). FLH restarts a copy of ONS in each round as an expert, and chooses the best one using expert-tracking algorithms. The meta-algorithm used to track the best expert is inspired by the Fixed-Share algorithm [Herbster and Warmuth, 1998]. While FLH is equipped with an $O(\frac{d}{\alpha} \log T)$ weakly adaptive regret, it is computationally expensive since it needs to maintain $t$ experts in the $t$-th iteration. To reduce the computational cost, Hazan and Seshadhri [2007] further prune the number of experts based on a data streaming algorithm. In this way, FLH only keeps $O(\log t)$ experts, at the price of an $O(\frac{d}{\alpha} \log^2 T)$ weakly adaptive regret. Notice that the efficient version of FLH essentially creates and removes experts dynamically. As pointed out by Adamskiy et al. [2012], this behavior can be modeled by the sleeping expert setting [Freund et al., 1997], in which the expert can be "asleep" for certain rounds and does not make any advice.

For general convex functions, we can use OGD as the expert-algorithm in FLH. Hazan and Seshadhri [2007] prove that FLH and its efficient variant attain $O(\sqrt{T \log T})$ and $O(\sqrt{T \log^3 T})$ weakly adaptive regrets, respectively. This result reveals a limitation of weakly adaptive regret—it does not respect short intervals well. For example, the $O(\sqrt{T \log T})$ regret bound is meaningless for intervals of length $O(\sqrt{T})$. To address this limitation, Daniely et al. [2015] introduce the strongly adaptive regret which takes the interval length as a parameter, as shown in (1). They propose a novel meta-algorithm, named as Strongly Adaptive Online Learner (SAOL). SAOL carefully constructs a set of intervals, then runs an instance of low-regret algorithm in each interval as an expert, and finally combines active experts' outputs by a variant of multiplicative weights method [Arora et al., 2012]. SAOL also maintains $O(\log t)$ experts in the $t$-th round, and achieves an $O(\sqrt{\tau} \log T)$ strongly adaptive regret for convex functions. Later, Jun et al. [2017a] develop a new meta-algorithm named as sleeping coin betting (SCB), and improve the strongly adaptive regret bound to $O(\sqrt{\tau \log T})$. Very recently, Cutkosky [2020] proposes a novel strongly adaptive method, which can guarantee the $O(\sqrt{\tau \log T})$ bound in the worst case, while achieving tighter results when the square norms of gradients are small.

For $\lambda$-strongly convex functions, Zhang et al. [2018b] point out that we can replace ONS with OGD, and obtain an $O(\frac{1}{\lambda} \log T)$ weakly adaptive regret. They also demonstrate that the number of active experts can be reduced from $t$ to $O(\log t)$, at a cost of an additional $\log T$ factor in the regret. All the aforementioned adaptive algorithms need to query the gradient of the loss function at least $\Theta(\log t)$ times in the $t$-th iteration. Based on surrogate losses, Wang et al. [2018] show that the number of gradient evaluations per round can be reduced to 1 without affecting the performance. When we have prior knowledge about the change of environments, it is also possible to improve the logarithmic factor in the adaptive regret [Wan et al., 2021].

### 2.3 Dynamic regret

Another metric for dealing with changing environments is dynamic regret, which compares the cumulative loss of the online learner against a sequence of decisions [Zinkevich, 2003, Jadbabaie et al., 2015, Besbes et al., 2015, Yang et al., 2016, Zhang et al., 2017, 2018a]. Although we can upper bound dynamic regret by adaptive regret, the results are usually suboptimal [Zhang et al., 2018b, 2020, Cutkosky, 2020, Baby and Wang, 2021]. For more discussions, please refer to Zhang [2020].

## 3 Main results

We first present necessary preliminaries, including assumptions, definitions and the technical challenge, then provide our universal algorithm and its theoretical guarantee.

### 3.1 Preliminaries

We introduce two common assumptions used in the study of OCO [Hazan, 2016].

**Assumption 1** *The diameter of the domain $\Omega$ is bounded by $D$, i.e.,*

$$\max_{\mathbf{x}, \mathbf{y} \in \Omega} \|\mathbf{x} - \mathbf{y}\| \leq D. \tag{2}$$

**Assumption 2** *The gradients of all the online functions are bounded by $G$, i.e.,*

$$\max_{\mathbf{w} \in \Omega} \|\nabla f_t(\mathbf{w})\| \leq G, \ \forall t \in [T]. \tag{3}$$

Next, we state definitions of strong convexity and exp-concavity [Boyd and Vandenberghe, 2004, Cesa-Bianchi and Lugosi, 2006].

**Definition 1** *A function $f : \Omega \mapsto \mathbb{R}$ is $\lambda$-strongly convex if*

$$f(\mathbf{y}) \geq f(\mathbf{x}) + \langle \nabla f(\mathbf{x}), \mathbf{y} - \mathbf{x} \rangle + \frac{\lambda}{2} \|\mathbf{y} - \mathbf{x}\|^2, \ \forall \mathbf{x}, \mathbf{y} \in \Omega.$$

**Definition 2** *A function $f : \Omega \mapsto \mathbb{R}$ is $\alpha$-exp-concave if $\exp(-\alpha f(\cdot))$ is concave over $\Omega$.*

The following property of exp-concave functions will be used later [Hazan et al., 2007, Lemma 3].

**Lemma 1** *For a function $f : \Omega \mapsto \mathbb{R}$, where $\Omega$ has diameter $D$, such that $\forall \mathbf{w} \in \Omega$, $\|\nabla f(\mathbf{w})\| \leq G$ and $\exp(-\alpha f(\cdot))$ is concave, the following holds for $\beta = \frac{1}{2} \min\{\frac{1}{4GD}, \alpha\}$:*

$$f(\mathbf{y}) \geq f(\mathbf{x}) + \langle \nabla f(\mathbf{x}), \mathbf{y} - \mathbf{x} \rangle + \frac{\beta}{2} \langle \nabla f(\mathbf{x}), \mathbf{y} - \mathbf{x} \rangle^2, \ \forall \mathbf{x}, \mathbf{y} \in \Omega.$$

#### 3.1.1 Technical challenge

Before introducing the proposed algorithm, we discuss the technical challenge of minimizing the adaptive regret of multiple types of convex functions simultaneously. Existing adaptive algorithms [Hazan and Seshadhri, 2007, Daniely et al., 2015, Jun et al., 2017a, Zhang et al., 2018b] share the same framework and contain 3 components:

- An expert-algorithm, which is able to minimize the static regret of a specific type of functions;

- A set of intervals, each of which is associated with an expert-algorithm that minimizes the regret of that interval;
- A meta-algorithm, which combines the predictions of active experts in each round.

To design a universal algorithm, a straightforward way is to use a universal method for static regret, such as MetaGrad [van Erven and Koolen, 2016], as the expert-algorithm. In this way, the expert-algorithm is able to handle the uncertainty of functions. However, the challenge lies in the design of the meta-algorithm, because the meta-algorithms used by previous studies also lack universality. For example, the meta-algorithm of Hazan and Seshadhri [2007] is able to deliver a tight meta-regret for exp-concave functions, but a loose one for general convex functions. Similarly, meta-algorithms of Daniely et al. [2015] and Jun et al. [2017a] incur at least $\Theta(\sqrt{\tau})$ meta-regret for intervals of length $\tau$, which is tolerable for convex functions but suboptimal for exp-concave functions.

To address the above challenge, we have found two different solutions.

1. In the first approach, we dig into MetaGrad and modify it to minimize the adaptive regret directly. MetaGrad itself is a two-layer algorithm, which runs multiple expert-algorithms, each with a different learning rate, and combines them with a meta-algorithm named as Tilted Exponentially Weighted Average (TEWA). To address the aforementioned challenge, we extend TEWA to support sleeping experts so that it can minimize the adaptive regret. The advantage of TEWA is that its meta-regret only depends on the number of experts instead of the length of the interval, e.g., Lemma 4 of van Erven and Koolen [2016], and thus does not affect the optimality of the regret.
2. In the second approach, we use an existing universal method (e.g., MetaGrad or Maler) as the expert-algorithm, and change the meta-algorithm. Inspired by a recent universal strategy for OCO [Zhang et al., 2021], we choose a meta-algorithm that enjoys second-order bounds with excess losses [Gaillard et al., 2014, Koolen and Erven, 2015, Wintenberger, 2017, Mhammedi et al., 2019], and use the *linearized* loss to measure the performance of experts. In this way, the meta-algorithm can exploit strong convexity and exponential concavity, and suffers a small meta-regret. While the first approach remains a two-layer algorithm, the second one is three-layer.

Due to the limitation of space, we only present the first approach in this conference paper, and will elaborate the second one in an extended version.

### 3.2 A parameter-free and adaptive algorithm for exp-concave functions

Recall that our goal is to design a universal algorithm for minimizing the adaptive regret of general convex, exp-concave, and strongly convex functions simultaneously. However, to facilitate understanding, we start with a simpler question: How to minimize the adaptive regret of exp-concave functions, without knowing the modulus of exp-concavity? By proposing a novel algorithm to answer the above question, we present the main techniques used in our paper. Then, we extend that algorithm to support other types of functions in the next section. Since our algorithm is built upon MetaGrad, we first review its key steps below.

#### 3.2.1 Review of MetaGrad

The reason that MetaGrad can minimize the regret of $\alpha$-exp-concave functions without knowing the value of $\alpha$ is because it enjoys a second-order regret bound:

$$\sum_{t=1}^{T} f_t(\mathbf{w}_t) - \sum_{t=1}^{T} f_t(\mathbf{w}) \leq \sum_{t=1}^{T} \langle \nabla f_t(\mathbf{w}_t), \mathbf{w}_t - \mathbf{w} \rangle = O\left(\sqrt{V_T d \log T} + d \log T\right) \quad (4)$$

where $V_T = \sum_{t=1}^{T} \langle \nabla f_t(\mathbf{w}_t), \mathbf{w}_t - \mathbf{w} \rangle^2$. Besides, Lemma 1 implies

$$\sum_{t=1}^{T} f_t(\mathbf{w}_t) - \sum_{t=1}^{T} f_t(\mathbf{w}) \leq \sum_{t=1}^{T} \langle \nabla f_t(\mathbf{w}_t), \mathbf{w}_t - \mathbf{w} \rangle - \frac{\beta}{2} \sum_{t=1}^{T} \langle \nabla f_t(\mathbf{w}_t), \mathbf{w}_t - \mathbf{w} \rangle^2. \quad (5)$$

Combining (4) with (5) and applying the AM-GM inequality, we immediately obtain

$$\sum_{t=1}^{T} f_t(\mathbf{w}_t) - \sum_{t=1}^{T} f_t(\mathbf{w}) = O\left(\frac{d}{\beta} \log T\right) = O\left(\frac{d}{\alpha} \log T\right).$$

```
t     1   2   3   4   5   6   7   8   9   10  11  12  13  14  15  16  17  ···
I_0 [ ][ ][ ][ ][ ][ ][ ][ ][ ][ ][ ][ ][ ][ ][ ][ ] ···
I_1   [     ][     ][     ][     ][     ][     ][     ] ···
I_2       [         ][         ][         ][         ] ···
I_3           [                 ][                 ] ···
I_4                   [                             ] ···
```

Figure 1: Geometric covering (GC) intervals of Daniely et al. [2015].

From the above discussion, it becomes clear that if we can establish a second-order regret bound for any interval $[p, q] \subseteq [T]$, we are able to minimize the adaptive regret even when $\alpha$ is unknown.

The way that MetaGrad attains the regret bound in (4) is to run a set of experts, each of which minimizes a surrogate loss parameterized by a learning rate $\eta$

$$\ell_t^\eta(\mathbf{w}) = -\eta\langle\nabla f_t(\mathbf{w}_t), \mathbf{w}_t - \mathbf{w}\rangle + \eta^2\langle\nabla f_t(\mathbf{w}_t), \mathbf{w}_t - \mathbf{w}\rangle^2 \tag{6}$$

and then combine the outputs of experts by a meta-algorithm named as Tilted Exponentially Weighted Average (TEWA). Specifically, it creates an expert $E^\eta$ for each $\eta$ in

$$\mathcal{S}(T) = \left\{ \frac{2^{-i}}{5DG} \,\middle|\, i = 0, 1, \ldots, \left\lceil \frac{1}{2}\log_2 T \right\rceil \right\} \tag{7}$$

and thus maintains $1 + \lceil\frac{1}{2}\log_2 T\rceil = O(\log T)$ experts during the learning process. By simultaneously considering multiple learning rates, MetaGrad is able to deal with the uncertainty of $V_T$. Since the surrogate loss $\ell_t^\eta(\cdot)$ is exp-concave, a variant of ONS is used as the expert-algorithm. Let $\mathbf{w}_t^\eta$ be the output of expert $E^\eta$ in the $t$-th round. MetaGrad calculates the final output $\mathbf{w}_t$ according to TEWA:

$$\mathbf{w}_t = \frac{\sum_\eta \pi_t^\eta \eta \mathbf{w}_t^\eta}{\sum_\eta \pi_t^\eta \eta} \tag{8}$$

where $\pi_t^\eta \propto \exp(-\sum_{i=1}^{t-1}\ell_i^\eta(\mathbf{w}_i^\eta))$.

### 3.2.2 Our approach

In this section, we discuss how to minimize the adaptive regret by extending MetaGrad. Following the idea of sleeping experts [Freund et al., 1997], the most straightforward way is to create $1 + \lceil\frac{1}{2}\log_2(q - p + 1)\rceil$ experts for each interval $[p, q] \in [T]$, and combine them with a meta-algorithm that supports sleeping experts. However, this simple approach is inefficient because the total number of experts is on the order of $O(T^2 \log T)$. To control the number of experts, we make use of the geometric covering (GC) intervals [Daniely et al., 2015] defined as

$$\mathcal{I} = \bigcup_{k \in \mathbb{N} \cup \{0\}} \mathcal{I}_k,$$

where

$$\mathcal{I}_k = \left\{ [i \cdot 2^k, (i+1) \cdot 2^k - 1] : i \in \mathbb{N} \right\}, \; k \in \mathbb{N} \cup \{0\}.$$

A graphical illustration of GC intervals is given in Fig. 1. We observe that each $\mathcal{I}_k$ is a partition of $\mathbb{N} \setminus \{1, \cdots, 2^k - 1\}$ to consecutive intervals of length $2^k$. The GC intervals can be generated on the fly, so we do not need to fix the horizon $T$. We note that similar intervals have been proposed by Veness et al. [2013].

Then, we only focus on intervals in $\mathcal{I}$. For each interval $I = [r, s] \in \mathcal{I}$, we will create $1 + \lceil\frac{1}{2}\log_2(s - r + 1)\rceil$ experts, each of which minimizes one surrogate loss in $\{\ell_t^\eta(\mathbf{w})|\eta \in \mathcal{S}(s - r + 1)\}$ during $I$, where $\ell_t^\eta(\cdot)$ and $\mathcal{S}(\cdot)$ are defined in (6) and (7), respectively. These experts become active in round $r$ and will be removed forever after round $s$. It is easy to verify that in each round $t$, the number of intervals that contain $t$ is $\lfloor\log_2 t\rfloor + 1$ [Daniely et al., 2015], and thus the number of active experts is at most

$$(\lfloor\log_2 t\rfloor + 1)\left(1 + \left\lceil\frac{1}{2}\log_2 t\right\rceil\right) = O(\log^2 t).$$

So, the number of active experts is larger than that of MetaGrad by a logarithmic factor, which is the price paid in computations for the adaptivity to every interval.

---
**Algorithm 1** A Parameter-free and Adaptive algorithm for Exp-concave functions (PAE)
---
1: $\mathcal{A}_0 = \emptyset$
2: **for** $t = 1$ **to** $T$ **do**
3:     **for all** $I \in \mathcal{I}$ that starts from $t$ **do**
4:         **for all** $\eta \in \mathcal{S}(|I|)$ **do**
5:             Create an expert $E_I^\eta$ by running the slave algorithm of MetaGrad to minimize $\ell_t^\eta(\cdot)$ during $I$, and set $L_{t-1,I}^\eta = 0$
6:             Add $E_I^\eta$ to the set of active experts: $\mathcal{A}_t = \mathcal{A}_{t-1} \cup \{E_I^\eta\}$
7:         **end for**
8:     **end for**
9:     Receive output $\mathbf{w}_{t,J}^\eta$ from each expert $E_J^\eta \in \mathcal{A}_t$
10:     Submit $\mathbf{w}_t$ in (9)
11:     Observe the loss $f_t(\cdot)$ and evaluate the gradient $\nabla f_t(\mathbf{w}_t)$
12:     **for all** $E_J^\eta \in \mathcal{A}_t$ **do**
13:         Update $L_{t,J}^\eta = L_{t-1,J}^\eta + \ell_t^\eta(\mathbf{w}_{t,J}^\eta)$
14:         Pass the surrogate loss $\ell_t^\eta(\cdot)$ to expert $E_J^\eta$
15:     **end for**
16:     Remove experts whose ending times are $t$ from $\mathcal{A}_t$
17: **end for**
---

Finally, we need to specify how to combine the outputs of active experts. From the construction of the surrogate loss in (6), we observe that each expert uses its own loss, which is very different from the traditional setting of prediction with expert advice in which all experts use the same loss. As a result, existing meta-algorithms for adaptive regret [Daniely et al., 2015, Jun et al., 2017a], which assume a fixed loss function, cannot be directly applied. In the literature, the setting that each expert uses a loss function that may be different from the loss functions used by the other experts has been studied by Chernov and Vovk [2009], who name it as prediction with expert evaluators' advice. Furthermore, they have proposed a general conversion to the sleeping expert case by only considering active rounds in the calculation of the cumulative losses. Following this idea, we extend TEWA to sleeping experts.

Our Parameter-free and Adaptive algorithm for Exp-concave functions (PAE) is summarized in Algorithm 1. In the $t$-th round, we first create an expert $E_I^\eta$ for each interval $I \in \mathcal{I}$ that starts from $t$ and each $\eta \in \mathcal{S}(|I|)$, where $\mathcal{S}(\cdot)$ is defined in (7), and introduce a variable $L_{t-1,I}^\eta$ to record the cumulative loss of $E_I^\eta$ (Step 5). The expert $E_I^\eta$ is the slave algorithm of MetaGrad [van Erven and Koolen, 2016] that minimizes $\ell_t^\eta(\cdot)$ during interval $I$. We also maintain a set $\mathcal{A}_t$ consisting of all the active experts (Step 6). Denote the prediction of expert $E_J^\eta$ at round $t$ as $\mathbf{w}_{t,J}^\eta$. In Step 9, PAE collects the predictions of all the active experts, and then submits the following solution in Step 10:

$$\mathbf{w}_t = \frac{1}{\sum_{E_J^\eta \in \mathcal{A}_t} \exp(-L_{t-1,J}^\eta)\eta} \sum_{E_J^\eta \in \mathcal{A}_t} \exp(-L_{t-1,J}^\eta)\eta \mathbf{w}_{t,J}^\eta. \qquad (9)$$

Compared with TEWA in (8), we observe that (9) focuses on active experts and ignores inactive ones. Although the extension is inspired by Chernov and Vovk [2009], our analysis is different because the surrogate losses take a special form of (6). Specifically, we exploit the special structure of losses and apply a simple inequality [Cesa-Bianchi et al., 2005, Lemma 1]. In this way, we do not need to introduce advanced concepts such as the mixability of Chernov and Vovk [2009]. The analysis is still challenging because of the dynamic change of active experts. In Step 11, PAE observes the loss $f_t(\cdot)$ and evaluates the gradient $\nabla f_t(\mathbf{w}_t)$ to construct the surrogate loss. In Step 13, it updates the cumulative loss of each active expert, and in Step 14 passes the surrogate loss to each expert such that it can make predictions for the next round. In Step 16, PAE removes experts whose ending times are $t$ from $\mathcal{A}_t$.

Due to limitations of space, we provide the expert-algorithm, as well as all the proofs, in the supplementary. The theoretical guarantee of PAE is given below.

**Theorem 1** *Under Assumptions 1 and 2, for any interval $[p, q] \subseteq [T]$ and any $\mathbf{w} \in \Omega$, PAE satisfies*

$$\sum_{t=p}^{q} \langle \nabla f_t(\mathbf{w}_t), \mathbf{w}_t - \mathbf{w} \rangle \leq 10DGa(p,q)b(p,q) + 3\sqrt{a(p,q)b(p,q)}\sqrt{\sum_{t=p}^{q} \langle \nabla f_t(\mathbf{w}_t), \mathbf{w}_t - \mathbf{w} \rangle^2}$$

*where*

$$a(p,q) = 2\log_2(2q) + \frac{1}{2} + \frac{d}{2}\ln\left(1 + \frac{2}{25d}(q-p+1)\right), \qquad (10)$$

$$b(p,q) = 2\lceil\log_2(q-p+2)\rceil. \qquad (11)$$

*Furthermore, if all the online functions are $\alpha$-exp-concave over interval $[p,q]$, we have*

$$\sum_{t=p}^{q} f_t(\mathbf{w}_t) - \sum_{t=p}^{q} f_t(\mathbf{w}) \leq \left(10DG + \frac{9}{2\beta}\right)a(p,q)b(p,q) = O\left(\frac{d\log q\log(q-p)}{\alpha}\right)$$

*where $\beta = \frac{1}{2}\min\{\frac{1}{4GD}, \alpha\}$.*

**Remark**   Theorem 1 indicates that PAE enjoys a second-order regret bound for any interval, which in turn implies a small regret for exp-concave functions. Specifically, for $\alpha$-exp-concave functions, PAE satisfies SA-Regret$(T, \tau) = O(\frac{d}{\alpha}\log\tau\log T)$, which matches the regret of efficient FLH [Hazan and Seshadhri, 2007]. This is a remarkable result given the fact that PAE is *agnostic* to $\alpha$.

### 3.3   A universal algorithm for minimizing the adaptive regret

In this section, we extend PAE to support strongly convex functions and general convex functions. Inspired by Wang et al. [2020], we introduce a new surrogate loss to handle strong convexity:

$$\hat{\ell}_t^{\eta}(\mathbf{w}) = -\eta\langle\nabla f_t(\mathbf{w}_t), \mathbf{w}_t - \mathbf{w}\rangle + \eta^2\|\nabla f_t(\mathbf{w}_t)\|^2\|\mathbf{w}_t - \mathbf{w}\|^2 \qquad (12)$$

which is also parameterized by $\eta > 0$. Our goal is to attain another second-order type of regret bound

$$\sum_{t=p}^{q} f_t(\mathbf{w}_t) - \sum_{t=p}^{q} f_t(\mathbf{w}) \leq \sum_{t=p}^{q}\langle\nabla f_t(\mathbf{w}_t), \mathbf{w}_t - \mathbf{w}\rangle = \widetilde{O}\left(\sqrt{\sum_{t=p}^{q}\|\mathbf{w}_t - \mathbf{w}\|^2}\right) \qquad (13)$$

for any interval $[p,q] \subseteq T$. Combining (13) with Definition 1, we can establish a tight regret bound for $\lambda$-strongly convex functions over any interval without knowing the value of $\lambda$. Furthermore, we can also derive a regret bound for general convex functions from (13), and there is no need to add additional surrogate losses.

Our Universal algorithm for Minimizing the Adaptive regret (UMA) is summarized in Algorithm 2. UMA is a natural extension of PAE by incorporating the new surrogate loss $\hat{\ell}_t^{\eta}(\cdot)$. The overall procedure of UMA is very similar to PAE, except that the number of experts doubles and the weighting formula is modified accordingly. Specifically, in each round $t$, we further create an expert $\widehat{E}_I^{\eta}$ for each interval $I \in \mathcal{I}$ that starts from $t$ and each $\eta \in \mathcal{S}(|I|)$. $\widehat{E}_I^{\eta}$ is an instance of AOGD [Bartlett et al., 2008] that is able to minimize $\hat{\ell}_t^{\eta}$ during interval $I$. We use $\widehat{L}_{t-1,I}^{\eta}$ to represent the cumulative loss of $\widehat{E}_I^{\eta}$ till round $t-1$, and $\widehat{\mathcal{A}}_t$ to store all the active $\widehat{E}_I^{\eta}$'s. Denote the prediction of expert $\widehat{E}_J^{\eta}$ at round $t$ as $\widehat{\mathbf{w}}_{t,J}^{\eta}$. In Step 11, UMA receives predictions from experts in $\mathcal{A}_t$ and $\widehat{\mathcal{A}}_t$, and submits the following solution in Step 12:

$$\mathbf{w}_t = \frac{\sum_{E_J^{\eta}\in\mathcal{A}_t}\exp(-L_{t-1,J}^{\eta})\eta\mathbf{w}_{t,J}^{\eta} + \sum_{\widehat{E}_J^{\eta}\in\widehat{\mathcal{A}}_t}\exp(-\widehat{L}_{t-1,J}^{\eta})\eta\widehat{\mathbf{w}}_{t,J}^{\eta}}{\sum_{E_J^{\eta}\in\mathcal{A}_t}\exp(-L_{t-1,J}^{\eta})\eta + \sum_{\widehat{E}_J^{\eta}\in\widehat{\mathcal{A}}_t}\exp(-\widehat{L}_{t-1,J}^{\eta})\eta} \qquad (14)$$

which is an extension of (9) to accommodate more experts.

Our analysis shows that UMA inherits the theoretical guarantee of PAE, and meanwhile is able to minimize the adaptive regret of general convex and strongly convex functions.

**Theorem 2**   *Under Assumptions 1 and 2, for any interval $[p,q] \subseteq [T]$ and any $\mathbf{w} \in \Omega$, UMA enjoys the theoretical guarantee of PAE in Theorem 1. Besides, it also satisfies*

$$\sum_{t=p}^{q}\langle\nabla f_t(\mathbf{w}_t), \mathbf{w}_t - \mathbf{w}\rangle \leq 10DG\hat{a}(p,q)b(p,q) + 3G\sqrt{\hat{a}(p,q)b(p,q)}\sqrt{\sum_{t=p}^{q}\|\mathbf{w}_t - \mathbf{w}\|^2}, \qquad (15)$$

$$\sum_{t=p}^{q}\langle\nabla f_t(\mathbf{w}_t), \mathbf{w}_t - \mathbf{w}\rangle \leq 10DG\hat{a}(p,q)b(p,q) + 21DG\sqrt{\hat{a}(p,q)(q-p+1)} \qquad (16)$$

**Algorithm 2** A Universal algorithm for Minimizing the Adaptive regret (UMA)

1: $\mathcal{A}_0 = \widehat{\mathcal{A}}_0 = \emptyset$
2: **for** $t = 1$ **to** $T$ **do**
3:     **for all** $I \in \mathcal{I}$ that starts from $t$ **do**
4:         **for all** $\eta \in \mathcal{S}(|I|)$ **do**
5:             Create an expert $E_I^\eta$ by running the slave algorithm of MetaGrad to minimize $\ell_t^\eta(\cdot)$ during $I$, and set $L_{t-1,I}^\eta = 0$
6:             Add $E_I^\eta$ to the set of active experts: $\mathcal{A}_t = \mathcal{A}_{t-1} \cup \{E_I^\eta\}$
7:             Create an expert $\widehat{E}_I^\eta$ by running an instance of AOGD to minimize $\hat{\ell}_t^\eta(\cdot)$ during $I$, and set $\widehat{L}_{t-1,I}^\eta = 0$
8:             Add $\widehat{E}_I^\eta$ to the set of active experts: $\widehat{\mathcal{A}}_t = \widehat{\mathcal{A}}_{t-1} \cup \{\widehat{E}_I^\eta\}$
9:         **end for**
10:     **end for**
11:     Receive output $\mathbf{w}_{t,J}^\eta$ from each expert $E_J^\eta \in \mathcal{A}_t$ and $\widehat{\mathbf{w}}_{t,J}^\eta$ from each expert $\widehat{E}_J^\eta \in \widehat{\mathcal{A}}_t$
12:     Submit $\mathbf{w}_t$ in (14)
13:     Observe the loss $f_t(\cdot)$ and evaluate the gradient $\nabla f_t(\mathbf{w}_t)$
14:     **for all** $E_J^\eta \in \mathcal{A}_t$ **do**
15:         Update $L_{t,J}^\eta = L_{t-1,J}^\eta + \ell_t^\eta(\mathbf{w}_{t,J}^\eta)$
16:         Pass the surrogate loss $\ell_t^\eta(\cdot)$ to expert $E_J^\eta$
17:     **end for**
18:     **for all** $\widehat{E}_J^\eta \in \widehat{\mathcal{A}}_t$ **do**
19:         Update $\widehat{L}_{t,J}^\eta = \widehat{L}_{t-1,J}^\eta + \hat{\ell}_t^\eta(\mathbf{w}_{t,J}^\eta)$
20:         Pass the surrogate loss $\hat{\ell}_t^\eta(\cdot)$ to expert $\widehat{E}_J^\eta$
21:     **end for**
22:     Remove experts whose ending times are $t$ from $\mathcal{A}_t$ and $\widehat{\mathcal{A}}_t$
23: **end for**

*where $b(\cdot, \cdot)$ is given in (11), and*

$$\hat{a}(p, q) = 1 + 2\log_2(2q) + \log(q - p + 2). \tag{17}$$

*Furthermore, if all the online functions are $\lambda$-strongly convex over interval $[p, q]$, we have*

$$\sum_{t=p}^{q} f_t(\mathbf{w}_t) - \sum_{t=p}^{q} f_t(\mathbf{w}) \leq \left(10DG + \frac{9G^2}{2\lambda}\right)\hat{a}(p,q)b(p,q) = O\left(\frac{\log q \log(q-p)}{\lambda}\right).$$

**Remark** First, (15) shows that UMA is equipped with another second-order regret bound for any interval, leading to a small regret for strongly convex functions. Specifically, for $\lambda$-strongly convex functions, UMA achieves $\text{SA-Regret}(T, \tau) = O(\frac{1}{\lambda}\log \tau \log T)$, which matches the regret of the efficient algorithm of Zhang et al. [2018b]. Second, (16) manifests that UMA attains an $O(\sqrt{\tau \log T})$ strongly adaptive regret for general convex functions, which again matches the state-of-the-art result of Jun et al. [2017a] exactly. Finally, because of the dual adaptivity, UMA can handle the tough case that the type of functions switches or the parameter of functions changes.

## 4 Experiments

We present some empirical results to evaluate the proposed UMA, and the full details can be found in the supplementary.

First, we focus on exp-concave functions, and perform online classification on the ijcnn1 dataset [Chang and Lin, 2011, Prokhorov, 2001], where the logistic loss is used. In each round, we sample a mini-batch of examples to construct the loss, and flip the labels per 200 rounds to simulate changing environments. We compare UMA with FLH for exp-concave functions (abbr. FLH$_{\text{exp}}$) [Hazan and Seshadhri, 2007], ONS [Hazan et al., 2007], SCB [Jun et al., 2017a], and MetaGrad [van Erven and Koolen, 2016]. Notice that (i) both FLH$_{\text{exp}}$ and ONS need to know the modulus of exp-concavity beforehand; and (ii) SCB can be applied because exp-concave functions are also convex. Second, we

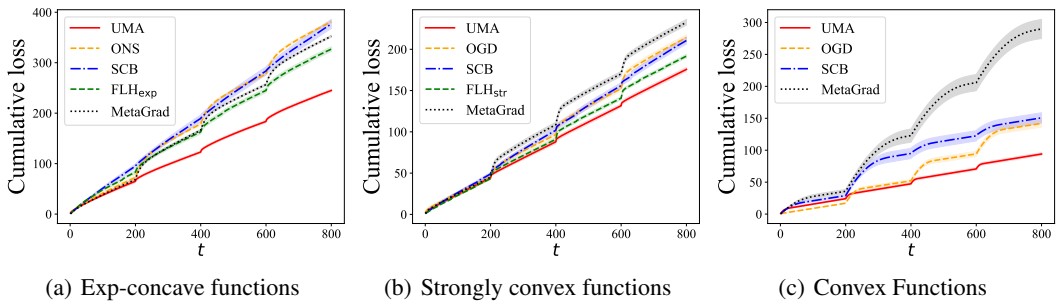

(a) Exp-concave functions  (b) Strongly convex functions  (c) Convex Functions

Figure 2: Cumulative losses of different methods versus the number of iterations.

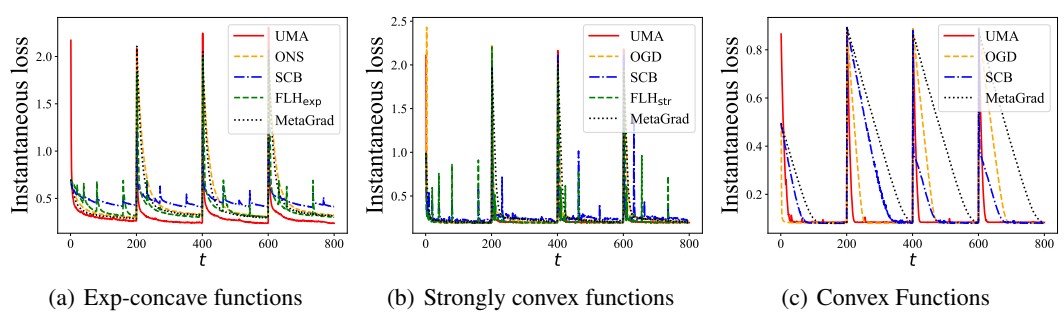

(a) Exp-concave functions  (b) Strongly convex functions  (c) Convex Functions

Figure 3: Instantaneous losses of different methods versus the number of iterations.

consider strongly convex functions, and follow the above setting but choose the regularized hinge loss. We compare UMA with FLH for strongly convex functions (abbr. FLH$_{str}$) [Zhang et al., 2018b], OGD [Shalev-Shwartz et al., 2007], SCB, and MetaGrad. Similarly, both FLH$_{str}$ and OGD require the modulus of strong convexity as their input. Finally, we investigate convex functions, and implement online linear regression on synthetic data, where the absolute loss is chosen. We generate the data in a way such that the underlying parameter changes every 200 rounds, and compare UMA with SCB, OGD [Zinkevich, 2003] and MetaGrad.

We conduct all the experiments on a personal laptop equipped with Intel i7-10750H CPU and 16G memory. We report the cumulative losses of different methods in Fig. 2. As can be seen, UMA performs nearly the best in all cases. Specifically, UMA is better than FLH$_{exp}$ in Fig. 2(a) and close to FLH$_{str}$ in Fig. 2(b), which indicates that UMA can estimate the moduli of exp-concavity and strong convexity automatically. We also observe that the cumulative losses of ONS, OGD, and MetaGrad, which are designed to minimize the static regret, increase rapidly over time, especially when the optimal model changes. We also show the average instantaneous losses of different methods in Fig. 3, and have similar observations.

## 5 Conclusion and future work

In this paper, we develop a universal algorithm that is able to minimize the adaptive regret of general convex, exp-concave and strongly convex functions simultaneously. For each type of functions, our theoretical guarantee matches the performance of existing algorithms specifically designed for this type of functions under apriori knowledge of parameters.

In the literature, it is well-known that smoothness can be exploited to improve the static regret for different types of loss functions [Srebro et al., 2010, Orabona et al., 2012, Wang et al., 2020]. Recent studies [Jun et al., 2017b, Zhang et al., 2019] have demonstrated that smoothness can also be exploited to improve the adaptive regret. In the future, we will investigate whether our universal algorithm for minimizing the adaptive regret can be extended to support smoothness.

## Acknowledgments and Disclosure of Funding

We are grateful for the valuable comments from anonymous reviewers and the area chair, especially regarding the second approach in Section 3.1.1. Funding in direct support of this work: NSFC grant 62122037 and 61976112, JiangsuSF grant BK20200064, Open Research Projects of Zhejiang Lab (NO. 2021KB0AB02).

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
