# Supplementary Material of "Dual Adaptivity: A Universal Algorithm for Minimizing the Adaptive Regret of Convex Functions"

**Lijun Zhang**[*], **Guanghui Wang**[*], **Wei-Wei Tu**[†], **Wei Jiang**[*], **Zhi-Hua Zhou**[*]

[*]National Key Laboratory for Novel Software Technology, Nanjing University, Nanjing, China
[†]4Paradigm Inc., Beijing 100000, China
{zhanglj, wanggh, jiangw, zhouzh}@lamda.nju.edu.cn, tuwwcn@gmail.com

## A  Algorithms for Experts

In this section, we provide the detail procedures of the two expert-algorithms in PAE and UMA.

### A.1  The first expert-algorithm: the slave algorithm of MetaGrad

We use the slave algorithm of MetaGrad to minimize $\ell_t^\eta(\cdot)$ during interval $I$.[1] We provide the procedure of expert $E_I^\eta$ in Algorithm 3. The generalized projection $\Pi_\Omega^A(\cdot)$ associated with a positive semidefinite matrix $A$ is defined as

$$\Pi_\Omega^A(\mathbf{x}) = \underset{\mathbf{w}\in\Omega}{\operatorname{argmin}}(\mathbf{w} - \mathbf{x})^\top A^{-1}(\mathbf{w} - \mathbf{x})$$

which is used in Step 5 of Algorithm 3.

---

**Algorithm 3** Expert $E_I^\eta$: Slave algorithm of MetaGrad

---

1: **Input:** Interval $I = [r, s]$, $\eta$
2: Let $\mathbf{w}_{r,I}^\eta$ be any point in $\Omega$ and $\Sigma_r^\eta = D^2 I$,
3: **for** $t = r$ **to** $s$ **do**
4:    Update

$$\Sigma_{t+1}^\eta = \Sigma_t^\eta - \frac{2\eta^2 \Sigma_t^\eta \mathbf{g}_t \mathbf{g}_t^\top \Sigma_t^\eta}{1 + 2\eta^2 \mathbf{g}_t^\top \Sigma_t^\eta \mathbf{g}_t}$$

   where

$$\mathbf{g}_t = \nabla f_t(\mathbf{w}_t)$$

5:    Calculate

$$\mathbf{w}_{t+1,I}^\eta = \Pi_\Omega^{\Sigma_{t+1}^\eta}\left(\mathbf{w}_t^\eta - \eta\Sigma_{t+1}^\eta \mathbf{g}_t\left(1 + 2\eta\mathbf{g}_t^\top\left(\mathbf{w}_t^\eta - \mathbf{w}_t\right)\right)\right)$$

6: **end for**

---

### A.2  The second expert-algorithm: Adaptive online gradient descent (AOGD)

It is easy to verify that the $\hat{\ell}_t^\eta(\cdot)$ in (12) enjoys the following property [Wang et al., 2020, Lemma 3 and Lemma 4].

---

[1]It is easy to verify that the surrogate loss $\ell_t^\eta(\cdot)$ in (6) is exp-concave [Wang et al., 2019], so we can also apply online Newton step (ONS) [Hazan et al., 2007] as an alternative.

35th Conference on Neural Information Processing Systems (NeurIPS 2021).

---

**Algorithm 4** Expert $\widehat{E}_I^\eta$: Adaptive Online Gradient Descent (AOGD)

---

1: **Input:** Interval $I = [r, s]$, $\eta$
2: Let $\widehat{\mathbf{w}}_{r,I}^\eta$ be any point in $\Omega$
3: **for** $t = r$ **to** $s$ **do**
4:     Update

$$\widehat{\mathbf{w}}_{t+1,I}^\eta = \Pi_\Omega \left( \widehat{\mathbf{w}}_{t,I}^\eta - \frac{1}{\alpha_t} \nabla \hat{\ell}_t^\eta(\widehat{\mathbf{w}}_{t,I}^\eta) \right)$$

where

$$\nabla \hat{\ell}_t^\eta(\widehat{\mathbf{w}}_{t,I}^\eta) = \eta \nabla f_t(\mathbf{w}_t) + 2\eta^2 \|\nabla f_t(\mathbf{w}_t)\|^2 (\widehat{\mathbf{w}}_{t,I}^\eta - \mathbf{w}_t)$$

$$\alpha_t = 2\eta^2 G^2 + 2\eta^2 \sum_{i=r}^{t} \|\nabla f_i(\mathbf{w}_i)\|^2$$

5: **end for**

---

**Lemma 2** *Under Assumptions 1 and 2, $\hat{\ell}_t^\eta(\cdot)$ in (12) is $2\eta^2 \|\nabla f_t(\mathbf{w}_t)\|^2$-strongly convex, and*

$$\max_{\mathbf{w} \in \Omega} \|\nabla \hat{\ell}_t^\eta(\mathbf{w})\| \le 2\eta^2 \|\nabla f_t(\mathbf{w}_t)\|^2, \ \forall \eta \le \frac{1}{5GD}.$$

Thus, although $\hat{\ell}_t^\eta(\cdot)$ is strongly convex, the modulus of strong convexity, i.e., $2\eta^2 \|\nabla f_t(\mathbf{w}_t)\|^2$ is not fixed. So, we choose AOGD [Bartlett et al., 2008] instead of OGD [Hazan et al., 2007] as the expert-algorithm to minimize $\hat{\ell}_t^\eta(\cdot)$ during interval $I$. We provide the procedure of expert $\widehat{E}_I^\eta$ in Algorithm 4. The projection operator $\Pi_\Omega(\cdot)$ is defined as

$$\Pi_\Omega(\mathbf{x}) = \underset{\mathbf{w} \in \Omega}{\operatorname{argmin}} \|\mathbf{w} - \mathbf{x}\|.$$

# B   Analysis

Here, we present proofs of main theorems.

## B.1   Proof of Theorem 1

We start with the meta-regret of PAE over any interval in $\mathcal{I}$.

**Lemma 3** *Under Assumptions 1 and 2, for any interval $I = [r, s] \in \mathcal{I}$ and any $\eta \in \mathcal{S}(s - r + 1)$, the meta-regret of PAE with respect to $E_I^\eta$ satisfies*

$$\sum_{t=r}^{s} \ell_t^\eta(\mathbf{w}_t) - \sum_{t=r}^{s} \ell_t^\eta(\mathbf{w}_{t,I}^\eta) = -\sum_{t=r}^{s} \ell_t^\eta(\mathbf{w}_{t,I}^\eta) \le 2 \log_2(2s).$$

The proof of Lemma 3 could be found in Appendix B.1.1. Combining Lemma 3 with the regret of expert $E_I^\eta$, which is just the regret bound of slave algorithm of MetaGrad over $I$, we establish a second-order regret of PAE over any interval in $\mathcal{I}$.

**Lemma 4** *Under Assumptions 1 and 2, for any interval $I = [r, s] \in \mathcal{I}$ and any $\mathbf{w} \in \Omega$, PAE satisfies*

$$\sum_{t=r}^{s} \langle \nabla f_t(\mathbf{w}_t), \mathbf{w}_t - \mathbf{w} \rangle \le 3 \sqrt{a(r,s) \sum_{t=r}^{s} \langle \nabla f_t(\mathbf{w}_t), \mathbf{w}_t - \mathbf{w} \rangle^2 + 10DGa(r,s)} \qquad (18)$$

*where $a(\cdot, \cdot)$ is defined in (10).*

The detailed proof of Lemma 4 is in Appendix B.1.2. To proceed, we introduce the following property of GC intervals [Daniely et al., 2015, Lemma 1.2].

**Lemma 5** *For any interval $[p, q] \subseteq [T]$, it can be partitioned into two sequences of disjoint and consecutive intervals, denoted by $I_{-m}, \ldots, I_0 \in \mathcal{I}$ and $I_1, \ldots, I_n \in \mathcal{I}$, such that*

$$|I_{-i}|/|I_{-i+1}| \leq 1/2, \ \forall i \geq 1$$

*and*

$$|I_i|/|I_{i-1}| \leq 1/2, \ \forall i \geq 2.$$

Based on the lemma above, we extend Lemma 4 to any interval $[p, q] \subseteq [T]$. Specifically, from Lemma 5, we conclude that $n \leq \lceil \log_2(q - p + 2) \rceil$ because otherwise

$$|I_1| + \cdots + |I_n| \geq 1 + 2 + \ldots + 2^{n-1} = 2^n - 1 > q - p + 1 = |I|.$$

Similarly, we have $m + 1 \leq \lceil \log_2(q - p + 2) \rceil$.

For any interval $[p, q] \subseteq [T]$, let $I_{-m}, \ldots, I_0 \in \mathcal{I}$ and $I_1, \ldots, I_n \in \mathcal{I}$ be the partition described in Lemma 5. Then, we have

$$\sum_{t=p}^{q} \langle \nabla f_t(\mathbf{w}_t), \mathbf{w}_t - \mathbf{w} \rangle = \sum_{i=-m}^{n} \sum_{t \in I_i} \langle \nabla f_t(\mathbf{w}_t), \mathbf{w}_t - \mathbf{w} \rangle. \tag{19}$$

Combining with Lemma 4, we have

$$\sum_{t=p}^{q} \langle \nabla f_t(\mathbf{w}_t), \mathbf{w}_t - \mathbf{w} \rangle$$

$$\leq \sum_{i=-m}^{n} \left( 3\sqrt{a(p,q) \sum_{t \in I_i} \langle \nabla f_t(\mathbf{w}_t), \mathbf{w}_t - \mathbf{w} \rangle^2} + 10DGa(p,q) \right)$$

$$= 10DG(m + 1 + n)a(p,q) + 3\sqrt{a(p,q)} \sum_{i=-m}^{n} \sqrt{\sum_{t \in I_i} \langle \nabla f_t(\mathbf{w}_t), \mathbf{w}_t - \mathbf{w} \rangle^2}$$

$$\leq 10DG(m + 1 + n)a(p,q) + 3\sqrt{(m + 1 + n)a(p,q)} \sqrt{\sum_{i=-m}^{n} \sum_{t \in I_i} \langle \nabla f_t(\mathbf{w}_t), \mathbf{w}_t - \mathbf{w} \rangle^2} \tag{20}$$

$$= 10DG(m + 1 + n)a(p,q) + 3\sqrt{(m + 1 + n)a(p,q)} \sqrt{\sum_{t=p}^{q} \langle \nabla f_t(\mathbf{w}_t), \mathbf{w}_t - \mathbf{w} \rangle^2}$$

$$\leq 10DGa(p,q)b(p,q) + 3\sqrt{a(p,q)b(p,q)} \sqrt{\sum_{t=p}^{q} \langle \nabla f_t(\mathbf{w}_t), \mathbf{w}_t - \mathbf{w} \rangle^2}.$$

When all the online functions are $\alpha$-exp-concave, Lemma 1 implies

$$\sum_{t=p}^{q} f_t(\mathbf{w}_t) - \sum_{t=p}^{q} f_t(\mathbf{w})$$

$$\leq \sum_{t=p}^{q} \langle \nabla f_t(\mathbf{w}_t), \mathbf{w}_t - \mathbf{w} \rangle - \frac{\beta}{2} \sum_{t=p}^{q} \langle \nabla f_t(\mathbf{w}_t), \mathbf{w}_t - \mathbf{w} \rangle^2$$

$$\overset{(20)}{\leq} 10DGa(p,q)b(p,q) + 3\sqrt{a(p,q)b(p,q)} \sqrt{\sum_{t=p}^{q} \langle \nabla f_t(\mathbf{w}_t), \mathbf{w}_t - \mathbf{w} \rangle^2}$$

$$- \frac{\beta}{2} \sum_{t=p}^{q} \langle \nabla f_t(\mathbf{w}_t), \mathbf{w}_t - \mathbf{w} \rangle^2$$

$$\leq \left( 10DG + \frac{9}{2\beta} \right) a(p,q)b(p,q).$$

### B.1.1 Proof of Lemma 3

This lemma is an extension of Lemma 4 of van Erven and Koolen [2016] to sleeping experts. We first introduce the following inequality [Cesa-Bianchi et al., 2005, Lemma 1].

**Lemma 6** *For all $z \geq -\frac{1}{2}$, $\ln(1+z) \geq z - z^2$.*

For any $\mathbf{w} \in \Omega$ and any $\eta \leq \frac{1}{5GD}$, we have

$$\eta\langle\nabla f_t(\mathbf{w}_t), \mathbf{w}_t - \mathbf{w}\rangle \geq -\eta\|\nabla f_t(\mathbf{w}_t)\|\|\mathbf{w}_t - \mathbf{w}\| \overset{(2),(3)}{\geq} -\frac{1}{5}.$$

Then, according to Lemma 6, we have

$$\begin{aligned}
\exp\left(-\ell_t^\eta(\mathbf{w})\right) &= \exp\left(\eta\langle\nabla f_t(\mathbf{w}_t), \mathbf{w}_t - \mathbf{w}\rangle - \eta^2\langle\nabla f_t(\mathbf{w}_t), \mathbf{w}_t - \mathbf{w}\rangle^2\right) \\
&\leq 1 + \eta\langle\nabla f_t(\mathbf{w}_t), \mathbf{w}_t - \mathbf{w}\rangle.
\end{aligned} \tag{21}$$

Recall that $\mathcal{A}_t$ is the set of active experts in round $t$, and $L_{t,J}^\eta$ is the cumulative loss of expert $E_J^\eta$. We have

$$\begin{aligned}
\sum_{E_J^\eta \in \mathcal{A}_t} \exp(-L_{t,J}^\eta) &= \sum_{E_J^\eta \in \mathcal{A}_t} \exp(-L_{t-1,J}^\eta)\exp\left(-\ell_t^\eta(\mathbf{w}_{t,J}^\eta)\right) \\
&\overset{(21)}{\leq} \sum_{E_J^\eta \in \mathcal{A}_t} \exp(-L_{t-1,J}^\eta)\left(1 + \eta\langle\nabla f_t(\mathbf{w}_t), \mathbf{w}_t - \mathbf{w}_{t,J}^\eta\rangle\right) \\
&= \sum_{E_J^\eta \in \mathcal{A}_t} \exp(-L_{t-1,J}^\eta) + \left\langle\nabla f_t(\mathbf{w}_t), \sum_{E_J^\eta \in \mathcal{A}_t} \exp(-L_{t-1,J}^\eta)\eta\mathbf{w}_t - \sum_{E_J^\eta \in \mathcal{A}_t} \exp(-L_{t-1,J}^\eta)\eta\mathbf{w}_{t,J}^\eta\right\rangle \\
&\overset{(9)}{=} \sum_{E_J^\eta \in \mathcal{A}_t} \exp(-L_{t-1,J}^\eta).
\end{aligned} \tag{22}$$

Summing (22) over $t = 1, \ldots, s$, we have

$$\sum_{t=1}^s \sum_{E_J^\eta \in \mathcal{A}_t} \exp(-L_{t,J}^\eta) \leq \sum_{t=1}^s \sum_{E_J^\eta \in \mathcal{A}_t} \exp(-L_{t-1,J}^\eta)$$

which can be rewritten as

$$\begin{aligned}
&\sum_{E_J^\eta \in \mathcal{A}_s} \exp(-L_{s,J}^\eta) + \sum_{t=1}^{s-1}\left(\sum_{E_J^\eta \in \mathcal{A}_t \setminus \mathcal{A}_{t+1}} \exp(-L_{t,J}^\eta) + \sum_{E_J^\eta \in \mathcal{A}_t \cap \mathcal{A}_{t+1}} \exp(-L_{t,J}^\eta)\right) \\
&\leq \sum_{E_J^\eta \in \mathcal{A}_1} \exp(-L_{0,J}^\eta) + \sum_{t=2}^s\left(\sum_{E_J^\eta \in \mathcal{A}_t \setminus \mathcal{A}_{t-1}} \exp(-L_{t-1,J}^\eta) + \sum_{E_J^\eta \in \mathcal{A}_t \cap \mathcal{A}_{t-1}} \exp(-L_{t-1,J}^\eta)\right)
\end{aligned}$$

implying

$$\begin{aligned}
&\sum_{E_J^\eta \in \mathcal{A}_s} \exp(-L_{s,J}^\eta) + \sum_{t=1}^{s-1} \sum_{E_J^\eta \in \mathcal{A}_t \setminus \mathcal{A}_{t+1}} \exp(-L_{t,J}^\eta) \\
&\leq \sum_{E_J^\eta \in \mathcal{A}_1} \exp(-L_{0,J}^\eta) + \sum_{t=2}^s \sum_{E_J^\eta \in \mathcal{A}_t \setminus \mathcal{A}_{t-1}} \exp(-L_{t-1,J}^\eta) \\
&= \sum_{E_J^\eta \in \mathcal{A}_1} \exp(0) + \sum_{t=2}^s \sum_{E_J^\eta \in \mathcal{A}_t \setminus \mathcal{A}_{t-1}} \exp(0) \\
&= |\mathcal{A}_1| + \sum_{t=2}^s |\mathcal{A}_t \setminus \mathcal{A}_{t-1}|.
\end{aligned} \tag{23}$$

Note that $|\mathcal{A}_1| + \sum_{t=2}^{s} |\mathcal{A}_t \setminus \mathcal{A}_{t-1}|$ is the total number of experts created until round $s$. From the structure of GC intervals and (7), we have

$$|\mathcal{A}_1| + \sum_{t=2}^{s} |\mathcal{A}_t \setminus \mathcal{A}_{t-1}| \leq s \left( \lfloor \log_2 s \rfloor + 1 \right) \left( 1 + \left\lceil \frac{1}{2} \log_2 s \right\rceil \right) \leq 4s^2. \tag{24}$$

From (23) and (24), we have

$$\sum_{E_J^\eta \in \mathcal{A}_s} \exp(-L_{s,J}^\eta) + \sum_{t=1}^{s-1} \sum_{E_J^\eta \in \mathcal{A}_t \setminus \mathcal{A}_{t+1}} \exp(-L_{t,J}^\eta) \leq 4s^2.$$

Thus, for any interval $I = [r, s] \in \mathcal{I}$, we have

$$\exp(-L_{s,I}^\eta) = \exp\left( -\sum_{t=r}^{s} \ell_t^\eta(\mathbf{w}_{t,I}^\eta) \right) \leq 4s^2$$

which completes the proof.

### B.1.2   Proof of Lemma 4

The analysis is similar to the proofs of Theorem 7 of van Erven and Koolen [2016] and Theorem 1 of Wang et al. [2019].

From Lemma 5 of van Erven and Koolen [2016], we have the following expert-regret of $E_I^\eta$.

**Lemma 7** *Under Assumptions 1 and 2, for any interval* $I = [r, s] \in \mathcal{I}$, *any* $\mathbf{w} \in \Omega$ *and any* $\eta \in \mathcal{S}(s - r + 1)$, *the expert-regret of* $E_I^\eta$ *satisfies*

$$\sum_{t=r}^{s} \ell_t^\eta(\mathbf{w}_{t,I}^\eta) - \sum_{t=r}^{s} \ell_t^\eta(\mathbf{w}) \leq \frac{\|\mathbf{w}_{r,I}^\eta - \mathbf{w}\|^2}{2D^2} + \frac{1}{2} \ln \det \left( I + 2\eta^2 D^2 \sum_{t=r}^{s} M_t \right),$$

where $M_t = \mathbf{g}_t \mathbf{g}_t^\top$ and $\mathbf{g}_t = \nabla f_t(\mathbf{w}_t)$. Based on Lemma 7, we have

$$\sum_{t=r}^{s} \ell_t^\eta(\mathbf{w}_{t,I}^\eta) - \sum_{t=r}^{s} \ell_t^\eta(\mathbf{w}) \overset{(2)}{\leq} \frac{1}{2} + \frac{1}{2} \sum_{i=1}^{d} \ln \left( 1 + 2\eta^2 D^2 \lambda_i \left( \sum_{t=r}^{s} \mathbf{g}_t \mathbf{g}_t^\top \right) \right)$$

$$\leq \frac{1}{2} + \frac{d}{2} \ln \left( 1 + \frac{2\eta^2 D^2}{d} \sum_{i=1}^{d} \lambda_i \left( \sum_{t=r}^{s} \mathbf{g}_t \mathbf{g}_t^\top \right) \right)$$

$$= \frac{1}{2} + \frac{d}{2} \ln \left( 1 + \frac{2\eta^2 D^2}{d} \text{tr} \left( \sum_{t=r}^{s} \mathbf{g}_t \mathbf{g}_t^\top \right) \right)$$

$$= \frac{1}{2} + \frac{d}{2} \ln \left( 1 + \frac{2\eta^2 D^2}{d} \sum_{t=r}^{s} \|g_t\|_2^2 \right)$$

$$\leq \frac{1}{2} + \frac{d}{2} \ln \left( 1 + \frac{2}{25d}(s - r + 1) \right)$$

where the second inequality is by the concavity of the function $\ln x$ and Jensen's inequality and the last inequality is due to $\eta \leq \frac{1}{5DG}$. Combining the regret bounds in Lemmas 3 and 7, we have

$$-\sum_{t=r}^{s} \ell_t^\eta(\mathbf{w}) = \eta \sum_{t=r}^{s} \langle \nabla f_t(\mathbf{w}_t), \mathbf{w}_t - \mathbf{w} \rangle - \eta^2 \sum_{t=r}^{s} \langle \nabla f_t(\mathbf{w}_t), \mathbf{w}_t - \mathbf{w} \rangle^2$$

$$\leq 2 \log_2(2s) + \frac{1}{2} + \frac{d}{2} \ln \left( 1 + \frac{2}{25d}(s - r + 1) \right)$$

for any $\eta \in \mathcal{S}(s - r + 1)$. Thus,

$$\sum_{t=r}^{s} \langle \nabla f_t(\mathbf{w}_t), \mathbf{w}_t - \mathbf{w} \rangle \leq \frac{2 \log_2(2s) + \frac{1}{2} + \frac{d}{2} \ln \left( 1 + \frac{2}{25d}(s - r + 1) \right)}{\eta} + \eta \sum_{t=r}^{s} \langle \nabla f_t(\mathbf{w}_t), \mathbf{w}_t - \mathbf{w} \rangle^2 \tag{25}$$

for any $\eta \in \mathcal{S}(s - r + 1)$.

Let $a(r, s) = 2 \log_2(2s) + \frac{1}{2} + \frac{d}{2} \ln \left( 1 + \frac{2}{25d}(s - r + 1) \right) \geq 2$. Note that the optimal $\eta_*$ that minimizes the R.H.S. of (25) is

$$\eta_* = \sqrt{\frac{a(r, s)}{\sum_{t=r}^{s} \langle \nabla f_t(\mathbf{w}_t), \mathbf{w}_t - \mathbf{w} \rangle^2}} \geq \frac{\sqrt{2}}{GD\sqrt{s - r + 1}}.$$

Recall that

$$\mathcal{S}(s - r + 1) = \left\{ \frac{2^{-i}}{5DG} \mid i = 0, 1, \dots, \left\lceil \frac{1}{2} \log_2(s - r + 1) \right\rceil \right\}.$$

If $\eta_* \leq \frac{1}{5DG}$, there must exist an $\eta \in \mathcal{S}(s - r + 1)$ such that

$$\eta \leq \eta_* \leq 2\eta.$$

Then, (25) implies

$$\sum_{t=r}^{s} \langle \nabla f_t(\mathbf{w}_t), \mathbf{w}_t - \mathbf{w} \rangle \leq 2 \frac{a(r, s)}{\eta_*} + \eta_* \sum_{t=r}^{s} \langle \nabla f_t(\mathbf{w}_t), \mathbf{w}_t - \mathbf{w} \rangle^2$$

$$= 3 \sqrt{a(r, s) \sum_{t=r}^{s} \langle \nabla f_t(\mathbf{w}_t), \mathbf{w}_t - \mathbf{w} \rangle^2}. \tag{26}$$

On the other hand, if $\eta_* \geq \frac{1}{5DG}$, we have

$$\sum_{t=r}^{s} \langle \nabla f_t(\mathbf{w}_t), \mathbf{w}_t - \mathbf{w} \rangle^2 \leq 25D^2G^2 a(r, s).$$

Then, (25) with $\eta = \frac{1}{5DG}$ implies

$$\sum_{t=r}^{s} \langle \nabla f_t(\mathbf{w}_t), \mathbf{w}_t - \mathbf{w} \rangle \leq 5DG a(r, s) + 5DG a(r, s) = 10DG a(r, s). \tag{27}$$

We complete the proof by combining (26) and (27).

## B.2    Proof of Theorem 2

We first show the meta-regret of UMA, which is similar to Lemma 3 of PAE.

**Lemma 8** *Under Assumptions 1 and 2, for any interval $I = [r, s] \in \mathcal{I}$ and any $\eta \in \mathcal{S}(s - r + 1)$, the meta-regret of UMA satisfies*

$$\sum_{t=r}^{s} \ell_t^{\eta}(\mathbf{w}_t) - \sum_{t=r}^{s} \ell_t^{\eta}(\mathbf{w}_{t,I}^{\eta}) = -\sum_{t=r}^{s} \ell_t^{\eta}(\mathbf{w}_{t,I}^{\eta}) \leq 2 \log_2(2s),$$

$$\sum_{t=r}^{s} \hat{\ell}_t^{\eta}(\mathbf{w}_t) - \sum_{t=r}^{s} \hat{\ell}_t^{\eta}(\widehat{\mathbf{w}}_{t,I}^{\eta}) = -\sum_{t=r}^{s} \hat{\ell}_t^{\eta}(\widehat{\mathbf{w}}_{t,I}^{\eta}) \leq 2 \log_2(2s).$$

The proof of Lemma 8 is provided in Appendix B.2.1. Combining with the expert-regret of $E_I^{\eta}$ and $\widehat{E}_I^{\eta}$, we prove the following second-order regret of UMA over any interval in $\mathcal{I}$, which is similar to Lemma 4 of PAE.

**Lemma 9** *Under Assumptions 1 and 2, for any interval $I = [r, s] \in \mathcal{I}$ and any $\mathbf{w} \in \Omega$, UMA satisfies*

$$\sum_{t=r}^{s} \langle \nabla f_t(\mathbf{w}_t), \mathbf{w}_t - \mathbf{w} \rangle \leq 3 \sqrt{a(r, s) \sum_{t=r}^{s} \langle \nabla f_t(\mathbf{w}_t), \mathbf{w}_t - \mathbf{w} \rangle^2} + 10DG a(r, s), \tag{28}$$

$$\sum_{t=r}^{s} \langle \nabla f_t(\mathbf{w}_t), \mathbf{w}_t - \mathbf{w} \rangle \leq 3G \sqrt{\hat{a}(r, s) \sum_{t=r}^{s} \|\mathbf{w}_t - \mathbf{w}\|^2} + 10DG \hat{a}(r, s) \tag{29}$$

*where $a(\cdot, \cdot)$ and $\hat{a}(\cdot, \cdot)$ are defined in (10) and (17), respectively.*

The detailed proof of Lemma 9 is shown in B.2.2. Based on the property of GC intervals [Daniely et al., 2015, Lemma 1.2], we extend Lemma 9 to any interval $[p, q] \subseteq [T]$, which implies Theorem 2. Notice that (28) is the same as (18), so Theorem 1 also holds for UMA. In the following, we prove (15) in a similar way. Combining (19) with (29), we have

$$
\begin{aligned}
&\sum_{t=p}^{q} \langle \nabla f_t(\mathbf{w}_t), \mathbf{w}_t - \mathbf{w} \rangle \\
&\leq \sum_{i=-m}^{n} \left( 3G \sqrt{\hat{a}(p,q) \sum_{t \in I_i} \|\mathbf{w}_t - \mathbf{w}\|^2} + 10DG\hat{a}(p,q) \right) \\
&= 10DG(m+1+n)\hat{a}(p,q) + 3G\sqrt{\hat{a}(p,q)} \sum_{i=-m}^{n} \sqrt{\sum_{t \in I_i} \|\mathbf{w}_t - \mathbf{w}\|^2} \\
&\leq 10DG(m+1+n)\hat{a}(p,q) + 3G\sqrt{(m+1+n)\hat{a}(p,q)} \sqrt{\sum_{i=-m}^{n} \sum_{t \in I_i} \|\mathbf{w}_t - \mathbf{w}\|^2} \\
&= 10DG(m+1+n)\hat{a}(p,q) + 3G\sqrt{(m+1+n)\hat{a}(p,q)} \sqrt{\sum_{t=p}^{q} \|\mathbf{w}_t - \mathbf{w}\|^2} \\
&\leq 10DG\hat{a}(p,q)b(p,q) + 3G\sqrt{\hat{a}(p,q)b(p,q)} \sqrt{\sum_{t=p}^{q} \|\mathbf{w}_t - \mathbf{w}\|^2}.
\end{aligned}
\tag{30}
$$

We proceed to prove (16). If we upper bound $\sum_{t=p}^{q} \|\mathbf{w}_t - \mathbf{w}\|^2$ in (15) by $D^2(q - p + 1)$, we arrive at

$$
\sum_{t=p}^{q} \langle \nabla f_t(\mathbf{w}_t), \mathbf{w}_t - \mathbf{w} \rangle \leq 10DG\hat{a}(p,q)b(p,q) + 3DG\sqrt{\hat{a}(p,q)b(p,q)} \sqrt{q - p + 1}
$$

which is worse than (16) by a $\sqrt{b(p,q)}$ factor. To avoid this factor, we use a different way to simplify (30):

$$
\begin{aligned}
&\sum_{t=p}^{q} \langle \nabla f_t(\mathbf{w}_t), \mathbf{w}_t - \mathbf{w} \rangle \\
&\leq \sum_{i=-m}^{n} \left( 3G \sqrt{\hat{a}(p,q) \sum_{t \in I_i} \|\mathbf{w}_t - \mathbf{w}\|^2} + 10DG\hat{a}(p,q) \right) \\
&= 10DG(m+1+n)\hat{a}(p,q) + 3G\sqrt{\hat{a}(p,q)} \sum_{i=-m}^{n} \sqrt{\sum_{t \in I_i} \|\mathbf{w}_t - \mathbf{w}\|^2} \\
&\leq 10\hat{a}(p,q)b(p,q) + 3DG\sqrt{\hat{a}(p,q)} \sum_{i=-m}^{n} \sqrt{|I_i|}.
\end{aligned}
\tag{31}
$$

Let $J = [p, q]$. According to Lemma 5, we have [Daniely et al., 2015, Theorem 1]

$$
\sum_{i=-m}^{n} \sqrt{|I_i|} \leq 2 \sum_{i=0}^{\infty} \sqrt{2^{-i}|J|} \leq \frac{2\sqrt{2}}{\sqrt{2}-1} \sqrt{|J|} \leq 7\sqrt{|J|} = 7\sqrt{q - p + 1}.
\tag{32}
$$

We get (16) by combining (31) and (32).

When all the online functions are $\lambda$-strongly convex, Definition 1 implies

$$\sum_{t=p}^{q} f_t(\mathbf{w}_t) - \sum_{t=p}^{q} f_t(\mathbf{w})$$

$$\leq \sum_{t=p}^{q} \langle \nabla f_t(\mathbf{w}_t), \mathbf{w}_t - \mathbf{w} \rangle - \frac{\lambda}{2} \sum_{t=p}^{q} \|\mathbf{w}_t - \mathbf{w}\|^2$$

$$\overset{(15)}{\leq} 10 D G \hat{a}(p,q) b(p,q) + 3 G \sqrt{\hat{a}(p,q) b(p,q)} \sqrt{\sum_{t=p}^{q} \|\mathbf{w}_t - \mathbf{w}\|^2} - \frac{\lambda}{2} \sum_{t=p}^{q} \|\mathbf{w}_t - \mathbf{w}\|^2$$

$$\leq \left( 10 D G + \frac{9 G^2}{2\lambda} \right) \hat{a}(p,q) b(p,q).$$

### B.2.1 Proof of Lemma 8

The analysis is similar to that of Lemma 3. We first demonstrate that (21) also holds for the new surrogate loss $\hat{\ell}_t^{\eta}(\cdot)$.

Notice that

$$\langle \nabla f_t(\mathbf{w}_t), \mathbf{w}_t - \mathbf{w} \rangle^2 \leq \|\nabla f_t(\mathbf{w}_t)\|^2 \|\mathbf{w}_t - \mathbf{w}\|^2. \tag{33}$$

As a result, we have

$$\exp\left(-\hat{\ell}_t^{\eta}(\mathbf{w})\right) = \exp\left(\eta \langle \nabla f_t(\mathbf{w}_t), \mathbf{w}_t - \mathbf{w} \rangle - \eta^2 \|\nabla f_t(\mathbf{w}_t)\|^2 \|\mathbf{w}_t - \mathbf{w}\|^2\right)$$

$$\overset{(33)}{\leq} \exp\left(\eta \langle \nabla f_t(\mathbf{w}_t), \mathbf{w}_t - \mathbf{w} \rangle - \eta^2 \langle \nabla f_t(\mathbf{w}_t), \mathbf{w}_t - \mathbf{w} \rangle^2\right) = \exp\left(-\ell_t^{\eta}(\mathbf{w})\right) \tag{34}$$

$$\overset{(21)}{\leq} 1 + \eta \langle \nabla f_t(\mathbf{w}_t), \mathbf{w}_t - \mathbf{w} \rangle$$

for any $\mathbf{w} \in \Omega$.

Then, we repeat the derivation of (22), and have

$$\sum_{E_J^{\eta} \in \mathcal{A}_t} \exp(-L_{t,J}^{\eta}) + \sum_{\widehat{E}_J^{\eta} \in \widehat{\mathcal{A}}_t} \exp(-\widehat{L}_{t,J}^{\eta})$$

$$= \sum_{E_J^{\eta} \in \mathcal{A}_t} \exp(-L_{t-1,J}^{\eta}) \exp\left(-\ell_t^{\eta}(\mathbf{w}_{t,J}^{\eta})\right) + \sum_{\widehat{E}_J^{\eta} \in \widehat{\mathcal{A}}_t} \exp(-\widehat{L}_{t-1,J}^{\eta}) \exp\left(-\hat{\ell}_t^{\eta}(\widehat{\mathbf{w}}_{t,J}^{\eta})\right)$$

$$\overset{(21),(34)}{\leq} \sum_{E_J^{\eta} \in \mathcal{A}_t} \exp(-L_{t-1,J}^{\eta}) \left(1 + \eta \langle \nabla f_t(\mathbf{w}_t), \mathbf{w}_t - \mathbf{w}_{t,J}^{\eta} \rangle\right)$$

$$+ \sum_{\widehat{E}_J^{\eta} \in \widehat{\mathcal{A}}_t} \exp(-\widehat{L}_{t-1,J}^{\eta}) \left(1 + \eta \langle \nabla f_t(\mathbf{w}_t), \mathbf{w}_t - \widehat{\mathbf{w}}_{t,J}^{\eta} \rangle\right)$$

$$= \sum_{E_J^{\eta} \in \mathcal{A}_t} \exp(-L_{t-1,J}^{\eta}) + \sum_{\widehat{E}_J^{\eta} \in \widehat{\mathcal{A}}_t} \exp(-\widehat{L}_{t-1,J}^{\eta})$$

$$+ \left\langle \nabla f_t(\mathbf{w}_t), \left( \sum_{E_J^{\eta} \in \mathcal{A}_t} \exp(-L_{t-1,J}^{\eta})\eta + \sum_{\widehat{E}_J^{\eta} \in \widehat{\mathcal{A}}_t} \exp(-\widehat{L}_{t-1,J}^{\eta})\eta \right) \mathbf{w}_t \right\rangle$$

$$- \left\langle \nabla f_t(\mathbf{w}_t), \sum_{E_J^{\eta} \in \mathcal{A}_t} \exp(-L_{t-1,J}^{\eta})\eta \mathbf{w}_{t,J}^{\eta} + \sum_{\widehat{E}_J^{\eta} \in \widehat{\mathcal{A}}_t} \exp(-\widehat{L}_{t-1,J}^{\eta})\eta \widehat{\mathbf{w}}_{t,J}^{\eta} \right\rangle$$

$$\overset{(14)}{=} \sum_{E_J^{\eta} \in \mathcal{A}_t} \exp(-L_{t-1,J}^{\eta}) + \sum_{\widehat{E}_J^{\eta} \in \widehat{\mathcal{A}}_t} \exp(-\widehat{L}_{t-1,J}^{\eta}).$$

Following the derivation of (23) and (24), we have

$$
\sum_{E_J^\eta \in \mathcal{A}_s} \exp(-L_{s,J}^\eta) + \sum_{t=1}^{s-1} \sum_{E_J^\eta \in \mathcal{A}_t \setminus \mathcal{A}_{t+1}} \exp(-L_{t,J}^\eta)
$$

$$
+ \sum_{\widehat{E}_J^\eta \in \widehat{\mathcal{A}}_s} \exp(-\widehat{L}_{s,J}^\eta) + \sum_{t=1}^{s-1} \sum_{\widehat{E}_J^\eta \in \widehat{\mathcal{A}}_t \setminus \widehat{\mathcal{A}}_{t+1}} \exp(-\widehat{L}_{t,J}^\eta)
$$

$$
\leq |\mathcal{A}_1| + \sum_{t=2}^{s} |\mathcal{A}_t \setminus \mathcal{A}_{t-1}| + |\widehat{\mathcal{A}}_1| + \sum_{t=2}^{s} |\widehat{\mathcal{A}}_t \setminus \widehat{\mathcal{A}}_{t-1}|
$$

$$
\leq 2s \left( \lfloor \log_2 s \rfloor + 1 \right) \left( 1 + \left\lceil \frac{1}{2} \log_2 s \right\rceil \right) \leq 4s^2.
$$

Thus, for any interval $I = [r, s] \in \mathcal{I}$, we have

$$
\exp(-L_{s,I}^\eta) = \exp \left( -\sum_{t=r}^{s} \ell_t^\eta(\mathbf{w}_{t,I}^\eta) \right) \leq 4s^2 \text{ and } \exp(-\widehat{L}_{s,I}^\eta) = \exp \left( -\sum_{t=r}^{s} \widehat{\ell}_t^\eta(\widehat{\mathbf{w}}_{t,I}^\eta) \right) \leq 4s^2
$$

which completes the proof.

### B.2.2 Proof of Lemma 9

First, (28) can be established by combining Lemmas 8 and 7, and following the proof of Lemma 4. Next, we prove (29) in a similar way.

From Lemma 2 and the property of AOGD [Bartlett et al., 2008], we have the following expert-regret of $\widehat{E}_I^\eta$ [Wang et al., 2020, Theorem 2].

**Lemma 10** *Under Assumptions 1 and 2, for any interval $I = [r, s] \in \mathcal{I}$ and any $\eta \in \mathcal{S}(s - r + 1)$, the expert-regret of $\widehat{E}_I^\eta$ satisfies*

$$
\sum_{t=r}^{s} \widehat{\ell}_t^\eta(\widehat{\mathbf{w}}_{t,I}^\eta) - \sum_{t=r}^{s} \widehat{\ell}_t^\eta(\mathbf{w}) \leq 1 + \log(s - r + 2), \ \forall \mathbf{w} \in \Omega.
$$

Combining the regret bound in Lemmas 8 and 10, we have

$$
-\sum_{t=r}^{s} \widehat{\ell}_t^\eta(\mathbf{w}) = \eta \sum_{t=r}^{s} \langle \nabla f_t(\mathbf{w}_t), \mathbf{w}_t - \mathbf{w} \rangle - \eta^2 \|f_t(\mathbf{w}_t)\|^2 \sum_{t=r}^{s} \|\mathbf{w}_t - \mathbf{w}\|^2
$$

$$
\leq 1 + 2\log_2(2s) + \log(s - r + 2)
$$

for any $\eta \in \mathcal{S}(s - r + 1)$. Thus,

$$
\sum_{t=r}^{s} \langle \nabla f_t(\mathbf{w}_t), \mathbf{w}_t - \mathbf{w} \rangle \leq \frac{1 + 2\log_2(2s) + \log(s - r + 2)}{\eta} + \eta \|\nabla f_t(\mathbf{w}_t)\|^2 \sum_{t=r}^{s} \|\mathbf{w}_t - \mathbf{w}\|^2
$$

$$
\overset{(3)}{\leq} \frac{1 + 2\log_2(2s) + \log(s - r + 2)}{\eta} + \eta G^2 \sum_{t=r}^{s} \|\mathbf{w}_t - \mathbf{w}\|^2
$$

(35)

for any $\eta \in \mathcal{S}(s - r + 1)$.

Let $\hat{a}(r, s) = 1 + 2\log_2(2s) + \log(s - r + 2) \geq 3$. Note that the optimal $\eta_*$ that minimizes the R.H.S. of (25) is

$$
\eta_* = \sqrt{\frac{\hat{a}(r, s)}{G^2 \sum_{t=r}^{s} \|\mathbf{w}_t - \mathbf{w}\|^2}} \geq \frac{\sqrt{3}}{GD\sqrt{s - r + 1}}.
$$

Recall that

$$
\mathcal{S}(s - r + 1) = \left\{ \frac{2^{-i}}{5DG} \ \middle| \ i = 0, 1, \ldots, \left\lceil \frac{1}{2} \log_2(s - r + 1) \right\rceil \right\}.
$$

If $\eta_* \leq \frac{1}{5DG}$, there must exist an $\eta \in \mathcal{S}(s - r + 1)$ such that

$$\eta \leq \eta_* \leq 2\eta.$$

Then, (35) implies

$$\sum_{t=r}^{s} \langle \nabla f_t(\mathbf{w}_t), \mathbf{w}_t - \mathbf{w} \rangle \leq 2\frac{\hat{a}(r,s)}{\eta_*} + \eta_* G^2 \sum_{t=r}^{s} \|\mathbf{w}_t - \mathbf{w}\|^2 = 3G \sqrt{\hat{a}(r,s) \sum_{t=r}^{s} \|\mathbf{w}_t - \mathbf{w}\|^2}. \quad (36)$$

On the other hand, if $\eta_* \geq \frac{1}{5DG}$, we have

$$\sum_{t=r}^{s} \|\mathbf{w}_t - \mathbf{w}\|^2 \leq 25D^2 \hat{a}(r,s).$$

Then, (35) with $\eta = \frac{1}{5DG}$ implies

$$\sum_{t=r}^{s} \langle \nabla f_t(\mathbf{w}_t), \mathbf{w}_t - \mathbf{w} \rangle \leq 5DG\hat{a}(r,s) + 5DG\hat{a}(r,s) = 10DG\hat{a}(r,s). \quad (37)$$

We obtain (29) by combining (36) and (37).

## C   Full experiments

In this section, we present the details of the experiments.[2]

### C.1   Experimental settings

First, we focus on exp-concave functions, and perform online classification on the ijcnn1 dataset from LIBSVM Data [Chang and Lin, 2011, Prokhorov, 2001]. In each round, a batch of training examples $\{(\mathbf{x}_{t,1}, y_{t,1}), \ldots, (\mathbf{x}_{t,n}, y_{t,n})\}$ are sampled randomly from the dataset, where $(\mathbf{x}_{t,i}, y_{t,i}) \in [-1,1]^d \times \{-1,1\}, i = 1, \ldots, n$. The online learner aims to predict a linear model $\mathbf{w}_t$ and then suffers a logistic loss:

$$f_t(\mathbf{w}_t) = \frac{1}{n} \sum_{i=1}^{n} \log\left(1 + \exp(-y_{t,i}\mathbf{w}_t^\top \mathbf{x}_{t,i})\right).$$

To simulate the changing environment, the labels of samples are flipped every 200 iterations. To satisfy Assumption 1, we define the domain $\Omega$ as a $d$-dimensional ball with radius 10, i.e., $\Omega = \{\mathbf{w} \in \mathbb{R}^d | \|\mathbf{w}\| \leq 10\}$. For this dataset, $d = 22$, and we set $n = 512$, $D = 20$, and $G = \sqrt{22}$. As mentioned in Section 4, we compare UMA with FLH for exp-concave functions (abbr. FLH$_{\text{exp}}$) [Hazan and Seshadhri, 2007], ONS [Hazan et al., 2007], SCB [Jun et al., 2017a] and MetaGrad [van Erven and Koolen, 2016]. Since both FLH$_{\text{exp}}$ and ONS need to know the modulus of exp-concavity beforehand, we use grid search to determine this parameter. We search its value from $\{1e2, 1e1, 1, 1e-1, 1e-2, 1e-3\}$ and pick the best one for each algorithm.

For strongly convex functions, we follow the above setting but choose the regularized hinge loss:

$$f_t(\mathbf{w}_t) = \frac{1}{n} \sum_{i=1}^{n} \max\left(0, 1 - y_{t,i}\mathbf{w}_t^\top \mathbf{x}_{t,i}\right) + \frac{\lambda}{2}\|\mathbf{w}_t\|^2.$$

In the experiments, we set $\lambda = 0.001$ and $G = (\sqrt{22} + 0.01)$. We then compare UMA with FLH for strongly convex functions (abbr. FLH$_{\text{str}}$) [Zhang et al., 2018b], OGD [Shalev-Shwartz et al., 2007], SCB and MetaGrad. Similarly, FLH$_{\text{str}}$ and OGD require the modulus of strong convexity as their input. We not only try the theoretically optimal value $\lambda = 0.001$, but also conduct grid search from $\{1e-1, 1e-2, 1e-3, 1e-4, 1e-5, 1e-6\}$. We then pick the best parameter for FLH$_{\text{str}}$ and OGD.

Finally, we investigate convex functions, and implement online linear regression on synthetic data. In each round $t$, a batch of data points $\{(\mathbf{x}_{t,1}, y_{t,1}), \ldots, (\mathbf{x}_{t,n}, y_{t,n})\}$ arrive, where $\mathbf{x}_{t,i} \in \mathbb{R}^d$ is

---

[2]The code is available from `https://github.com/Dual-Adaptivity/code`.

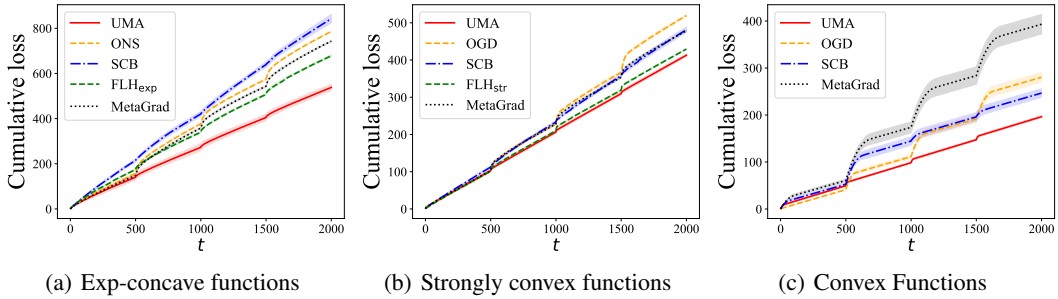

Figure 4: Cumulative losses of different methods when the interval size is 500.

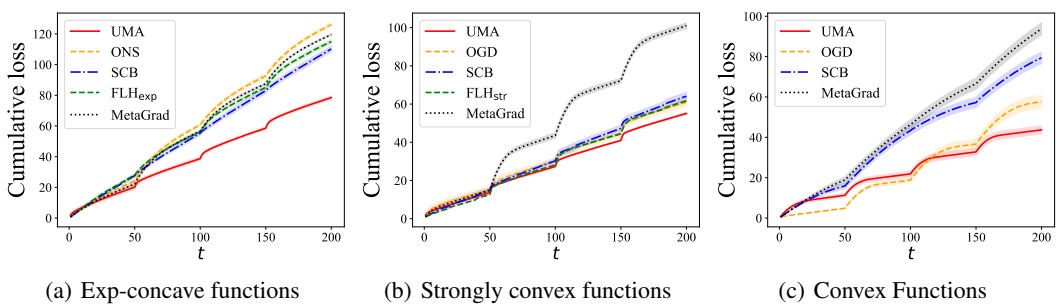

Figure 5: Cumulative losses of different methods when the interval size is 50.

sampled randomly inside a $d$-dimensional ball with radius 10. The target value $y_{t,i}$ is generated by $y_{t,i} = \mathbf{w}^\top \mathbf{x}_{t,i} + \epsilon$ where $\epsilon \sim \mathcal{N}(0, 0.01)$ is a zero-mean Gaussian noise with standard deviation 0.01. The unknown parameter $\mathbf{w}$ is sampled randomly from $[0,1]^d$ and $[-1,0]^d$ alternately every 200 rounds to evaluate the adaptivity of different methods. After predicting $\mathbf{w}_t$, the online learner suffers the absolute loss:

$$f_t(\mathbf{w}_t) = \frac{1}{n} \sum_{i=1}^{n} \left| \mathbf{w}_t^\top \mathbf{x}_{t,i} - y_{t,i} \right|.$$

In the experiments, we set $n = 512$, $d = 50$, $D = 20$, and $G = 10$, and compare UMA with SCB, OGD [Zinkevich, 2003] and MetaGrad. For OGD, we set the step size as $\frac{c}{\sqrt{t}}$. To decide the parameter $c$, we try the theoretically optimal $c = \frac{D}{G}$, together with grid search from $\{1e2, 1e1, 1, 1e-1, 1e-2, 1e-3\} \times \frac{D}{G}$, and use the one that leads to the best perfromance.

## C.2 Experimental results

We repeat each experiment 100 times, plot the average cumulative loss in Fig. 2, and report the average instantaneous loss in Fig. 3. We have the following observations.

- As can be seen, UMA can deal with different types of functions and performs nearly the best in all cases. Specifically, UMA is better than FLH$_{exp}$ in Fig. 2(a) and Fig. 3(a), and close to FLH$_{str}$ in Fig. 2(b) and Fig. 3(b), which indicates that UMA can estimate the moduli of exp-concavity and strong convexity automatically. So, UMA is adaptive to the type of functions and the nature of environments simultaneously.
- When facing changing environments, the three adaptive methods (UMA, FLH$_{exp}$ and FLH$_{str}$) perform better than the traditional algorithms (OGD, ONS and MetaGrad) designed for static regret. In Fig. 2, we observe that although OGD, ONS and MetaGrad perform well at the beginning, their cumulative losses increase rapidly when the optimal model changes. In Fig. 3, we can see that the instantaneous losses of all methods jump at rounds 200, 400 and 600, but the losses of adaptive methods decrease more rapidly.

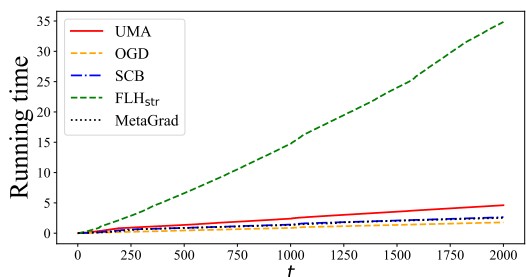

Figure 6: Running times of different methods.

We also test the case that the environment changes more quickly or slowly. Fig. 4 and Fig. 5 show the results when the underlying model changes every 500 rounds and 50 rounds, respectively. We observe that UMA still performs nearly the best in all cases, indicating its performance is stable across different interval sizes.

Finally, we record the running times of different methods in Fig. 6, which corresponds to the experiment in Fig. 4(b), i.e., strongly convex functions with interval size 500. OGD has a constant complexity per round, and its running time increases linearly. MetaGrad keeps $O(\log t)$ experts in the $t$-th round, and thus its running time is higher than that of OGD. We note that the two adaptive methods (SCB and FLH$_{\text{str}}$) not only maintain $O(\log t)$ experts, but also query the gradient $O(\log t)$ times. To reveal the effect of gradient evaluations, we use the variant of SCB based on surrogate losses [Wang et al., 2018], which only calculates the gradient once per round. As a result, the running time of SCB is close to that of MetaGrad, but the running time of FLH$_{\text{str}}$ is much longer than that of MetaGrad. Our UMA algorithm creates $O(\log^2 t)$ experts to ensure dual adaptivity, so it is slower than MetaGrad and SCB. On the other hand, UMA is faster than FLH$_{\text{str}}$ because it only evaluates the gradient once in each iteration.

## D  Supporting Lemmas

For the sake of completeness, we provide the proofs of Lemmas 2 and 10, which can be found in the full paper of Wang et al. [2020].

### D.1  Proof of Lemma 2

First, we show that

$$\hat{\ell}_t^\eta(\mathbf{y}) \geq \hat{\ell}_t^\eta(\mathbf{x}) + \langle \nabla \hat{\ell}_t^\eta(\mathbf{x}), \mathbf{y} - \mathbf{x} \rangle + \frac{2\eta^2 \|\nabla f_t(\mathbf{w}_t)\|^2}{2} \|\mathbf{y} - \mathbf{x}\|^2$$

for any $\mathbf{x}, \mathbf{y} \in \Omega$. When $\|\nabla f_t(\mathbf{w}_t)\| \neq 0$, it is easy to verify that $\hat{\ell}_t^\eta(\cdot)$ is $2\eta^2 \|\nabla f_t(\mathbf{w}_t)\|^2$-strongly convex, and the above inequality holds according to Definition 1. When $\|\nabla f_t(\mathbf{w}_t)\| = 0$, then by the definition of $\hat{\ell}_t^\eta(\cdot)$ in (12), we have

$$\hat{\ell}_t^\eta(\mathbf{w}) = \nabla \hat{\ell}_t^\eta(\mathbf{w}) = 2\eta^2 \|\nabla f_t(\mathbf{w}_t)\|^2 = 0$$

for any $\mathbf{w} \in \Omega$, and thus the inequality still holds.

Next, we upper bound the gradient of $\hat{\ell}_t^\eta(\cdot)$ as follows:

$$\begin{aligned}
\|\nabla \hat{\ell}_t^\eta(\mathbf{w})\|^2 &= \langle \eta \nabla f_t(\mathbf{w}_t) + 2\eta^2 \|\nabla f_t(\mathbf{w}_t)\|^2 (\mathbf{w} - \mathbf{w}_t), \eta \nabla f_t(\mathbf{w}_t) + 2\eta^2 \|\nabla f_t(\mathbf{w}_t)\|^2 (\mathbf{w} - \mathbf{w}_t) \rangle \\
&= \eta^2 \|\nabla f_t(\mathbf{w}_t)\|^2 + 4\eta^3 \|\nabla f_t(\mathbf{w}_t)\|^2 \langle \nabla f_t(\mathbf{w}_t), \mathbf{w} - \mathbf{w}_t \rangle + 4\eta^4 \|\nabla f_t(\mathbf{w}_t)\|^4 \|\mathbf{w} - \mathbf{w}_t\|^2 \\
&\overset{(2),(3)}{\leq} \eta^2 \|\nabla f_t(\mathbf{w}_t)\|^2 + \frac{4}{5}\eta^2 \|\nabla f_t(\mathbf{w}_t)\|^2 + \frac{4}{25}\eta^2 \|\nabla f_t(\mathbf{w}_t)\|^2 \\
&\leq 2\eta^2 \|\nabla f_t(\mathbf{w}_t)\|^2.
\end{aligned}$$

## D.2  Proof of Lemma 10

Let $\widehat{\mathbf{w}}_{t+1,I}^{\eta'} = \widehat{\mathbf{w}}_{t,I}^{\eta} - \frac{1}{\alpha_t}\nabla\hat{\ell}_t^{\eta}(\widehat{\mathbf{w}}_{t,I}^{\eta})$. By Lemma 2, we have

$$\hat{\ell}_t^{\eta}(\widehat{\mathbf{w}}_{t,I}^{\eta}) - \hat{\ell}_t^{\eta}(\mathbf{w}) \le \langle\nabla\hat{\ell}_t^{\eta}(\widehat{\mathbf{w}}_{t,I}^{\eta}), \widehat{\mathbf{w}}_{t,I}^{\eta} - \mathbf{w}\rangle - \frac{2\eta^2\|\nabla f_t(\mathbf{w}_t)\|^2}{2}\|\widehat{\mathbf{w}}_{t,I}^{\eta} - \mathbf{w}\|^2$$

$$= \alpha_t\langle\widehat{\mathbf{w}}_{t,I}^{\eta} - \widehat{\mathbf{w}}_{t+1,I}^{\eta'}, \widehat{\mathbf{w}}_{t,I}^{\eta} - \mathbf{w}\rangle - \frac{2\eta^2\|\nabla f_t(\mathbf{w}_t)\|^2}{2}\|\widehat{\mathbf{w}}_{t,I}^{\eta} - \mathbf{w}\|^2$$

for any $\mathbf{w} \in \Omega$. For the first term, we have

$$\langle\widehat{\mathbf{w}}_{t,I}^{\eta} - \widehat{\mathbf{w}}_{t+1,I}^{\eta'}, \widehat{\mathbf{w}}_{t,I}^{\eta} - \mathbf{w}\rangle$$

$$= \|\widehat{\mathbf{w}}_{t,I}^{\eta} - \mathbf{w}\|^2 + \langle\mathbf{w} - \widehat{\mathbf{w}}_{t+1,I}^{\eta'}, \widehat{\mathbf{w}}_{t,I}^{\eta} - \mathbf{w}\rangle$$

$$= \|\widehat{\mathbf{w}}_{t,I}^{\eta} - \mathbf{w}\|^2 - \|\widehat{\mathbf{w}}_{t+1,I}^{\eta'} - \mathbf{w}\|^2 - \langle\widehat{\mathbf{w}}_{t,I}^{\eta} - \widehat{\mathbf{w}}_{t+1,I}^{\eta'}, \widehat{\mathbf{w}}_{t+1,I}^{\eta'} - \mathbf{w}\rangle$$

$$= \|\widehat{\mathbf{w}}_{t,I}^{\eta} - \mathbf{w}\|^2 - \|\widehat{\mathbf{w}}_{t+1,I}^{\eta'} - \mathbf{w}\|^2 + \|\widehat{\mathbf{w}}_{t+1,I}^{\eta'} - \widehat{\mathbf{w}}_{t,I}^{\eta}\|^2 + \langle\widehat{\mathbf{w}}_{t+1,I}^{\eta'} - \widehat{\mathbf{w}}_{t,I}^{\eta}, \widehat{\mathbf{w}}_{t,I}^{\eta} - \mathbf{w}\rangle$$

which implies that

$$\langle\widehat{\mathbf{w}}_{t,I}^{\eta} - \widehat{\mathbf{w}}_{t+1,I}^{\eta'}, \widehat{\mathbf{w}}_{t,I}^{\eta} - \mathbf{w}\rangle = \frac{1}{2}\left(\|\widehat{\mathbf{w}}_{t,I}^{\eta} - \mathbf{w}\|^2 - \|\widehat{\mathbf{w}}_{t+1,I}^{\eta'} - \mathbf{w}\|^2 + \|\widehat{\mathbf{w}}_{t,I}^{\eta} - \widehat{\mathbf{w}}_{t+1,I}^{\eta'}\|^2\right)$$

and thus

$$\hat{\ell}_t^{\eta}(\widehat{\mathbf{w}}_{t,I}^{\eta}) - \hat{\ell}_t^{\eta}(\mathbf{w}) \le \frac{\alpha_t}{2}(\|\widehat{\mathbf{w}}_{t,I}^{\eta} - \mathbf{w}\|^2 - \|\widehat{\mathbf{w}}_{t+1,I}^{\eta'} - \mathbf{w}\|^2)$$

$$+ \frac{1}{2\alpha_t}\|\nabla\hat{\ell}_t^{\eta}(\widehat{\mathbf{w}}_{t,I}^{\eta})\|^2 - \frac{2\eta^2\|\nabla f_t(\mathbf{w}_t)\|^2}{2}\|\widehat{\mathbf{w}}_{t,I}^{\eta} - \mathbf{w}\|^2.$$

Summing up over $t = r$ to $s$, we have

$$\sum_{t=r}^{s}\hat{\ell}_t^{\eta}(\widehat{\mathbf{w}}_{t,I}^{\eta}) - \sum_{t=r}^{s}\hat{\ell}(\mathbf{w})$$

$$\le \frac{\alpha_r}{2}\|\widehat{\mathbf{w}}_{r,I}^{\eta} - \mathbf{w}\|^2 + \sum_{t=r}^{s}\left(\alpha_t - \alpha_{t-1} - 2\eta^2\|\nabla f_t(\mathbf{w}_t)\|^2\right)\frac{\|\widehat{\mathbf{w}}_{t,I}^{\eta} - \mathbf{w}\|^2}{2} + \frac{1}{2}\sum_{t=r}^{s}\frac{1}{\alpha_t}\|\nabla\hat{\ell}_t^{\eta}(\widehat{\mathbf{w}}_{r,I}^{\eta})\|^2$$

$$\le 1 + \frac{1}{2}\sum_{t=r}^{s}\frac{1}{\alpha_t}\|\nabla\hat{\ell}_t^{\eta}(\widehat{\mathbf{w}}_{r,I}^{\eta})\|^2 \le 1 + \frac{1}{2}\sum_{t=r}^{s}\frac{\|\nabla f_t(\mathbf{w}_t)\|^2}{G^2 + \sum_{i=r}^{t}\|\nabla f_t(\mathbf{w}_t)\|^2} \le 1 + \log(r - s + 1)$$

where the second inequality is due to the fact that $\alpha_t - \alpha_{t-1} - 2\eta^2\|\nabla f_t(\mathbf{w}_t)\|^2 = 0$ and $\eta \le \frac{1}{5DG}$, the third inequality is derived from Lemma 2, and the last inequality is due to the following lemma (when $d = 1$) [Hazan et al., 2007, Lemma 11].

**Lemma 11** *For $t = 1, \ldots, T$, let $\mathbf{u}_t \in \mathbb{R}^d$ be a sequence of vectors such that for some $r > 0$, $\|\mathbf{u}_t\| \le r$. Define $V_t = \sum_{t=1}^{T}\mathbf{u}_t\mathbf{u}_t^{\top} + \epsilon I$. Then*

$$\sum_{t=1}^{T}\|\mathbf{u}_t\|_{V_t^{-1}}^2 \le d\log\left(\frac{r^2T}{\epsilon} + 1\right).$$