# OpenReview forum: "Dual Adaptivity: A Universal Algorithm for Minimizing the Adaptive Regret of Convex Functions"
_NeurIPS.cc/2021/Conference — NeurIPS 2021 Poster_

### Official Review · Reviewer_x3iJ · 2021-07-13

**Rating:** 7
**Confidence:** 3

**Summary:**

In this papers, the authors tackle the problem of online optimization with unknown (or even changing) convex function types, which can be plain convex, exp-concave, or strongly convex. Their main goal is to design an algorithm that suffers asymptotically optimal regret on any time interval.

To this end they extend the MetaGrad algorithm, a method that guarantees low static regret against plain convex or exp-concave functions, to also handle strongly convex functions and in a way that guarantees low adaptive regret, not only static regret. Their main technical contribution is showing that this can be done by incorporating sleeping experts in the meta algorithm of MetaGrad.

**Limitations And Societal Impact:**

Mostly yes, I've only raised some concerns related to the experiments.

**Main Review:**

# Strengths and Weaknesses

## Strengths
- The main paper is well written and clearly poses the question and the challenges involved, what can be a bit hard in this case since there are so many moving pieces;
- Introducing the simpler version first that handles only exp-concave functions does help in the readability;
- Although no technique in isolation is novel, the clever combination of many different techniques to create a truly "universal algorithm" is a solid contribution.

## Weakenesses
- The proofs in the supplementary material are not very well organized and written.
- The experiments are not very well explained (more details later);

# Details and Questions for the Authors

The paper was a nice read and is a solid contribution. As I mentioned, no single piece in isolation is novel, but this is a good combination of many different techniques to devise an interesting "universal" algorithm. Two points I'd like the authors' thoughts are on the supplementary material writing and on the experiments.

The supplementary material has all the proofs, but while I tried to read it (skipping some parts since our time for review is limited), I often got lost. The problem is that the authors start by giving a brief overview of the proof of the main theorems, deferring the proof of a couple of lemmas for later. Yet, they never explicitly say they are going to do that, what makes the reader wonder if they simply omitted the proof or if the proofs are "well-known". Also, each subsection usually consists of one proof, but they often state the lemmas they will need before actually starting the proof. In the end, it is hard to figure out exactly where the statement of lemmas and discussion ends, and where the proofs actually begin.

About the experiments, there are some aspects of them that do not seem fair (or at least not well explained):
- The classical static algorithms (OGD and ONS) should have their parameters chosen by some kind of grid search. It is not clear from the text if the parameters were chosen to be the "theoretically optimal" or if the authors did pick the best parameters for OGD and ONS;
- Why is the interval set at 200? Changing the interval size changes the results by a lot? If not, discussing this (even if in the appendix) would be interesting;
- Different datasets change the results by a lot? You should at least give some reason for using only one single dataset;
- Since you are comparing with ONS and OGD, would it also make sense to compare with one of the adaptive algorithms for static regret (Metagrad or Maler)? Why none of these algorithms were used?
- It would be interesting to provide timing data and the laptop specs, if possible;


**Time Spent Reviewing:**

4

---

> ### Author Response · Authors · 2021-08-07
> **We will improve the writing and provide more experiments.**
>
> Many thanks for the constructive reviews!
>
> ---
>
> Answer 1: We greatly appreciate the comments on the proofs. Following the suggestions, we will improve our writing and address all the concerns.
>
> ---
>
> Answer 2: We will conduct more experiments and provide more details.
> - For OGD, we set its step-size as the "theoretically optimal" one, which works well. For ONS, we find that the "theoretically optimal" step-size preforms badly, and then choose its step-size by grid search.\
> According to your suggestion, recently we have tried to set the step-size of OGD by grid search, and obtained similar results. We will update our paper by using the grad search for both methods.
> - We have tested different interval sizes, and observe similar behaviors. We will provide more results with different interval sizes.
> - The results are quite stable across different datasets, and we will report more results.
> - We omit the universal algorithms for static regret (Metagrad or Maler) in the experiments, because their performance is very close to ONS or OGD. We would like to compare with them in the revised version.
> - We will provide timing data and the laptop specs in the revised version. Specifically, the laptop is equipped with Intel i7-10750H CPU and 16G memory.

---

> > ### Comment · Reviewer_x3iJ · 2021-08-31
> > **Comments based on reviewer discussion**
> >
> > First of all, thanks for the authors for the extra-information. We had some comments among reviewers that might be worthwhile commenting with you, hopefully some of these may be useful and incorporated in the final version of the paper.
> >
> > I have to say that the discussion (mostly the discussion with the AC and with jKjC) helped me understand a bit more the nuances of the problem, even more so as an "outsider" to this specific line of work, in particular the "3-layer algorithm" based on the recent Universal OCO algorithm. Their work is mostly concurrent to yours, so this does not affect in anyway novelty of your work, but we agree that discussing this "black-box" approach is very insightful (is it only more complicated, or is it less efficient as well? Are the guarantees the same?), and if added to the paper would be of enormous value. At some point one of the reviewers also mentioned adding in the experiments this 3-layer algorithm... this might be hard to do given the time-frame, and again, this work is concomitant with yours, but we were curious if it would perform as well as the approach used in the paper or if it would be worse (or at least less computationally efficient).
> >
> > Another question we had was whether the adaptation of TEWA to sleeping experts question only worked for the experts used in the paper of if it was more general. In the latter case, you could even add it in the abstract/introduction as a independent contribution in its own right. However, I went a bit through the meta-regret proof to see if it actually works for general sleeping experts, and it seems it does not. The authors do use that the active experts come from the GC intervals structure (line 462-463), and use through the whole proof a bit of structure of the loss functions, if I'm not mistaken (eq (21)). Maybe the proof can be adapted to a more general regret bound to sleeping experts, the authors might have a better answer regarding this point. Again, if the analysis does not hold for general sleeping experts, this is not a problem. Yet, if the analysis does hold for general sleeping expert, this should certainly be more explicitly stated (both in the intro and in theorem in the appendix), because at this point this is not clear.

---

> > > ### Author Response · Authors · 2021-09-01
> > > **Thanks for the information!**
> > >
> > > Dear Reviewer x3iJ,
> > >
> > > Many thanks for your comments!
> > >
> > > We will add more discussions about the “black-box" approach. Specifically, we plan to write an extended version which contains both the black-box approach (a 3-layer algorithm) and the current approach (a 2-layer algorithm).
> > >
> > > Unfortunately, the analysis does not hold for general sleeping experts. As you mentioned, (21) is the key inequality used in the proof. The adaptation of TEWA to sleeping experts only works for the experts that satisfies (21).
> > >
> > >
> > > Best\
> > > Authors

---

### Official Review · Reviewer_jQzg · 2021-07-14

**Rating:** 7
**Confidence:** 4

**Summary:**

In this paper, the authors develop a universal algorithm, namely UMA, that is able to minimize the adaptive regret of general convex, exp-concave and strongly convex functions simultaneously. The key idea is to modify MetaGrad to support sleeping experts and introduce additional surrogate losses for strongly convex functions and general convex functions. In this way, UMA can minimize the adaptive regret of multiple types of convex functions automatically, and thus is universal. Some experiments are done to evaluate the effectiveness of UMA.

**Limitations And Societal Impact:**

Questions:
1. All the adaptive regret bounds have a logarithmic dependence on $T$, which means that the regret over fixed-length intervals keeps increasing as $T$ increases. The logarithmic dependence is usually acceptable, but when $T$ becomes very large, this could be an issue. Is there any way to avoid the dependence on $T$?

2. In Line 50, the authors claim that UMA attains *second-order* regret bounds over any interval. On the other hand, Cutkosky [2020] also establishes a *second-order* bound for adaptive regret. Can the authors discuss the difference between them?


**Main Review:**

From the technical viewpoint, UMA is a nice extension of MetaGrad to minimize the adaptive regret. Because MetaGrad has a small meta-regret, the authors are able to maintain the optimality of UMA for different types of functions. This result actually demonstrates that the meta-algorithm of MetaGrad can be used to replace existing meta-algorithms designed for adaptive regret [Hazan and Seshadhri, 2007, Daniely et al., 2015, Jun et al., 2017a].

Strengths:
1. To the best of my knowledge, UMA is the first universal algorithm for minimizing the adaptive regret. Due to the property of universality, UMA can be applied with less manual intervention.
2. For each type of functions, UMA achieves state-of-the-art adaptive regret, and can also handle the case that the type of functions may change.
3. The paper is well-written and the proofs are easy to follow.

Weaknesses:
1. As pointed out by the authors, the main limitation of UMA is that the number of experts is $O(\log^2 t)$, which is larger than previous method by a logarithmic factor.
2. UMA is built upon MetaGrad, and somehow incremental.


**Time Spent Reviewing:**

4 hours

---

> ### Author Response · Authors · 2021-08-07
> **Many thanks for the constructive reviews!**
>
> Many thanks for the constructive reviews!
>
> ---
>
> Answer 1: The reason of the logarithmic dependence on $T$ is because we want to bound the regret over all the possible intervals in $[1,T]$. If we have some prior knowledge about the change of environments, it is possible to avoid the dependence on $\log T$. Specifically, consider the case that we have a lower bound $\tau_1$ and an upper bound $\tau_2$ on how long the environment changes. Then, we only need to focus on intervals with length in $[\tau_1, \tau_2]$, and can replace $\log T$ with $\log (\tau_2/\tau_1)$. We will add more discussions about this issue in the revised paper.
>
> ---
> Answer 2: The second-order bound of Cutkosky [2020] is very different from our second-order bounds. While the bound of Cutkosky can adapt to the squared norm of the gradients, it is inconsistent with the second-order properties of exp-concave functions and strongly convex functions. As a result, it cannot exploit exp-concavity and strong convexity, and we cannot use Cutkosky’s bound to design universal algorithms for OCO.

---

### Official Review · Reviewer_jKJC · 2021-07-16

**Rating:** 6
**Confidence:** 4

**Summary:**

This paper considers the problem of minimizing the adaptive regret together with being adaptive to the regularity of the loss function (exp-concavity, strong convexity, convexity,...). On one hand, algorithms and techniques to minimize the adaptive regret already exist but existing results are specific to the curvature of the losses. On the other hand, there exist universal algorithms that are adaptive to broad classes of loss functions (Metagrad and extensions). This paper combines both techniques and provides two algorithms (PAE and UMA) that get both guarantees simultaneously. In particular, it is adaptive to changes in the regularity of the losses. Finally, the authors provide some experiments to illustrate their results.

**Limitations And Societal Impact:**

Yes

**Main Review:**

On one hand, the paper is generally well-written, and having a generic algorithm that automatically adapts to the regularity of the losses to enjoy the best possible guarantees and can deal with non-stationary environment is surely interesting for the community and can be very useful for applications. In particular, I like the result to be adaptive to changes in the regularity of the loss sequences.

Yet, on the other hand, the results did not surprise me and the contribution is somewhat incremental in my opinion. The technique of sleeping experts is widely used and known to transform any static algorithm into an adaptive regret algorithm (see among many others, Daniely et al. 15, Orabona et al. 2017, Zhang et al. 2018). The Metagrad algorithm and its variants are based on combining experts. Combining their analysis with the sleeping expert procedure seems fairly straightforward and does not provide any technical novelty. In case I missed a new technical difficulty in the analysis, I encourage the authors to highlight it.

Moreover, I suspect that the results can be also obtained as follows. Second-order bounding algorithms that adapt to loss regularity already exist for the expert framework (see, for example, [1,2,3]). Using such algorithms as the meta-procedure for FLH (Zhang et al. 2018) with MetaGrad as subroutines should achieve this, doesn't it?


Another comment:
Another natural measure to deal with a non-stationary environment in online learning is dynamic regret. I do think that dynamic regret should be discussed in the introduction. As shown by Zhang et al 2018, there exists a reduction from adaptive regret to dynamic regret. Yet, it was shown that the rate obtained by Zhang et al. 2018 was suboptimal for exp-concave loss functions [4]. I would be surprised, but it would be interesting to see if the optimal rate of [4] could be recovered by your method for dynamic regret.


[1] A second-order bound with excess losses. Gaillard, Stoltz, Van Erven, 2014.
[2] Second-order quantile methods for experts and combinatorial games. Koolen, Van Erven, 2015.
[3] Optimal learning with Bernstein online aggregation. Wintenberger, 2017.
[4] Optimal Dynamic Regret in Exp-Concave Online Learning. D Baby, YX Wang, 2021.

**Time Spent Reviewing:**

2

---

> ### Author Response · Authors · 2021-08-07
> **Many thanks for the constructive reviews!**
>
> Many thanks for the constructive reviews! We hope the reviewer could reevaluate our paper, and are very happy to respond more questions during the rolling discussion.
>
> ---
> Q1: The technique of sleeping experts is widely used … The Metagrad algorithm and its variants are based on combining experts. Combining their analysis with the sleeping expert procedure seems fairly straightforward and does not provide any technical novelty.
>
> A1: In our opinion, the statement of “Combining their analysis with the sleeping expert procedure seems fairly straightforward and does not provide any technical novelty” underestimates our contributions.
> 1) In fact, all the existing adaptive algorithms for convex functions [Hazan and Seshadhri, 2007, Daniely et al., 2015, Jun et al., 2017a, Zhang et al., 2018] are combinations of sleeping experts with different expert-tracking algorithms. For example, Hazan and Seshadhri [2007] combine Fixed-Share with sleeping experts, and Jun et al. [2017a] combine coin betting with sleeping experts. The challenge is to find an appropriate expert-tracking algorithm for the problem at hand, and provide rigorous analysis when sleeping experts are present. So, such combinations are generally regarded as technically novel.
> 2) We would like to highlight our two contributions: (i) proposing to establish second-order bounds for convex functions on every interval; and (ii) demonstrating that the meta-algorithm of MetaGrad (i.e., TEWA) can be extended to support sleeping experts and then achieve our goal.
>
> ---
> Q2: In case I missed a new technical difficulty in the analysis, I encourage the authors to highlight it.
>
> A2: The technical difficulty is the extension of TEWA to sleeping experts, which requires careful analysis. During the proof of the meta-regret in the supplementary material, we need to struggle with the dynamic change of active experts. To verify the challenge, the reviewer could check the proof of Lemma 3, especially between (22) and (23).
>
> ---
> Q3: Second-order bounding algorithms that adapt to loss regularity already exist for the expert framework (see, for example, [1,2,3]). Using such algorithms as the meta-procedure for FLH (Zhang et al. 2018) with MetaGrad as subroutines should achieve this, doesn't it?
>
> A3: We cannot directly use the second-order algorithm of [1,2,3] as the meta-procedure for FLH. Because in this way, the upper bound will contains terms like $[f_t(\mathbf{w}_t)-f_t(\mathbf{w}_t^i)]^2$, where $\mathbf{w}_t^i$ is the output of the $i$-th expert, and it is unclear how to handle these terms.
>
> On the other hand, we do figure out a different universal algorithm for minimizing the adaptive regret of convex functions, which is similar to what the reviewer suggested.
> - We use the second-order algorithm in [1,2,3] as the meta-algorithm, but process the *linearized* loss instead of the original loss. This step is inspired by a recent work for universal OCO [6]. As shown in [6], in this way, we can exploit both exp-concavity and strong convexity.
> - We run MetaGrad as the expert-algorithm to process the *original* loss.
>
> However, the existence of the above algorithm does not affect the significance of our paper, as explained below.
> 1) It is essentially a more *complicated* version of our PAE algorithm by stacking one additional layer. Specifically, the above algorithm is a three-layer algorithm: the first layer is the second-order algorithm of [1,2,3], the middle layer is the meta-algorithm of MetaGrad, and the last layer is the expert-algorithm of MetaGrad. In contrast, our PAE is a two-layer algorithm: the first layer is the meta-algorithm of MetaGrad, and the second layer is the expert-algorithm of MetaGrad.
> 2) The key idea remains to establish second-order bounds over each interval, which is the contribution of our paper. The above algorithm is just a more complicated way to achieve this goal.
>
> [6] Zhang et al. A Simple yet Universal Strategy for Online Convex Optimization, arXiv:2105.03681, 2021.
>
> ---
> Q4: I do think that dynamic regret should be discussed in the introduction. As shown by Zhang et al 2018, there exists a reduction from adaptive regret to dynamic regret. Yet, it was shown that the rate obtained by Zhang et al. 2018 was suboptimal for exp-concave loss functions [4].
>
> A4: Thanks for the suggestion, and we will add more discussions about the dynamic regret. This paper aims to design a universal algorithm for adaptive regret of convex functions, instead of improving the existing adaptive regret bounds. So, our upper bounds are on same order as previous methods [Hazan and Seshadhri, 2007, Daniely et al., 2015, Jun et al., 2017a]. Following the reduction from adaptive regret to dynamic regret, we obtain the same bounds as Zhang et al. [2018].

---

> > ### Comment · Reviewer_jKJC · 2021-08-19
> > **Quick response to the authors**
> >
> > I thank the authors for their detailed response to my review. While I am currently travelling (for the next 1.5 week) and in a very tight time schedule, I only give a high level response. I also read other reviews.
> >
> > I believe that the authors did a good job in answering my concerns and after reading the other reviews I believe that I underestimated the contribution of the paper. I am happy to increase my score.

---

> > > ### Author Response · Authors · 2021-08-19
> > > **Many thanks! We will improve our paper accordingly.**
> > >
> > > Dear Reviewer jKJC,
> > >
> > > Thank you very much for your kind reply! We will revise our paper according to the constructive reviews.
> > >
> > > Best\
> > > Authors

---

### Official Review · Reviewer_Mr6y · 2021-07-26

**Rating:** 7
**Confidence:** 4

**Summary:**

This paper studies online convex optimization in the strongly-adaptive regret setting. Essentially, this paper generalizes MetaGrad, an OCO algorithm which was able to adapt to the type of convexity of the losses, to the strongly adaptive regret setting.

This algorithm is build using the metagrad meta-regret structure, geometric covering, a new surrogate loss function, and a generalization of the sleeping experts analysis. By deriving second-order regret bounds on the sub-intervals, the authors are able to show strongly-adaptive regret guarantees. The proposed algorithms were also shown to have favorable performance in experiments.


**Limitations And Societal Impact:**

yes

**Main Review:**

I think this is a good paper. It solves a natural problem -- the extension of universal OCO algorithms to strongly-adaptive regret -- and presents the new ideas in a clear manner. Generally, the explanations and writing are clear. The results are believable, and the analysis is sufficiently clear.

A few questions and suggestions for improvement:
1) Can you provide some intuition why the alternative surrogate function is needed?
2) Why spend so much time on PAE, as it is a special case of your main algorithm? Some important discussion, like the role and regret guarantees of the slave algorithms, could be a better use of the space in the main body.
3) Can you say anything about lower bounds?

post-rebuttal: Thanks for the clarification, and nice work! I remain positive about the paper.

**Time Spent Reviewing:**

3

---

> ### Author Response · Authors · 2021-08-07
> **Many thanks for the constructive reviews!**
>
> Many thanks for the constructive reviews!
>
> ---
> Answer 1: That is because exp-concave functions and strongly convex functions have *different* second-order properties, as shown in Lemma 1 and Definition 1. Based on the surrogate loss in (6), we derive the second-order regret bound in Theorem 1, which can utilize the second-order property of exp-concave functions. To future support strongly convex functions, we introduce the surrogate loss in (12) to obtain another second-order regret bound in Theorem 2, which can exploit the second-order property of strongly convex functions.
>
> ---
> Answer 2: Thanks for the comments. We just want to facilitate understanding, so we start with a simpler version. According to the suggestion, we will reorganize the paper to include more discussions.
>
> ---
> Answer 3: We can use the lower bounds of static regret to verify the optimality of our upper bounds. The minmax regret of general convex, $\alpha$-exp-concave and $\lambda$-strongly convex functions over intervals of length $\tau$ are $O(\sqrt{\tau})$, $O(\frac{d}{\alpha} \log \tau)$ and $O(\frac{1}{\lambda}\log \tau)$, respectively. So, our upper bounds for adaptive regret are optimal, up to logarithmic factors.

---

### Decision · Program_Chairs · 2021-09-27

**Decision:**

Accept (Poster)

**Comment:**

In agreement with the reviewers, I am happy to recommend acceptance of the paper. During the review process the authors promised several improvements to the paper (such as discussing and experimenting with the suggested simple black-box algorithm or clarifying that their references to sleeping experts are meant only in a restricted manner), which they must include in the final version of the paper.